# Microbiome disturbance and resilience dynamics of the upper respiratory tract during influenza A virus infection

Drishti Kaul[1,12], Raveen Rathnasinghe[2,12], Marcela Ferres[2], Gene S. Tan[1,3], Aldo Barrera[2,4], Brett E. Pickett[5,6], Barbara A. Methe[5,7], Suman Das[5], Isolda Budnik[2], Rebecca A. Halpin[5], David Wentworth[5,10], Mirco Schmolke [8,11], Ignacio Mena [8], Randy A. Albrecht [8], Indresh Singh[5], Karen E. Nelson[5], Adolfo García-Sastre [8,9], Chris L. Dupont [1✉] & Rafael A. Medina [2,4,8✉]

Infection with influenza can be aggravated by bacterial co-infections, which often results in disease exacerbation. The effects of influenza infection on the upper respiratory tract (URT) microbiome are largely unknown. Here, we report a longitudinal study to assess the temporal dynamics of the URT microbiomes of uninfected and influenza virus-infected humans and ferrets. Uninfected human patients and ferret URT microbiomes have stable healthy ecostate communities both within and between individuals. In contrast, infected patients and ferrets exhibit large changes in bacterial community composition over time and between individuals. The unhealthy ecostates of infected individuals progress towards the healthy ecostate, coinciding with viral clearance and recovery. *Pseudomonadales* associate statistically with the disturbed microbiomes of infected individuals. The dynamic and resilient microbiome during influenza virus infection in multiple hosts provides a compelling rationale for the maintenance of the microbiome homeostasis as a potential therapeutic target to prevent IAV associated bacterial co-infections.

[1] J. Craig Venter Institute, 4120 Capricorn Lane, La Jolla, CA 92037, USA. [2] Departmento de Enfermedades Infecciosas e Inmunología Pediátrica, Facultad de Medicina, Pontificia Universidad Católica de Chile, Santiago, Chile. [3] Department of Infectious Diseases, University of California San Diego, La Jolla, CA 92037, USA. [4] Millennium Institute on Immunology and Immunotherapy, Santiago, Chile. [5] J. Craig Venter Institute, 9704 Medical Center Drive, Rockville, MD 20850, 14, USA. [6] Microbiology & Molecular Biology, Brigham Young University, Provo, UT, USA. [7] Department of Medicine, University of Pittsburgh, Pittsburgh, PA 15213, USA. [8] Department of Microbiology, Global Health and Emerging Pathogens Institute, Icahn School of Medicine at Mount Sinai, New York, NY 10029, USA. [9] Department of Medicine, Icahn School of Medicine at Mount Sinai, New York, NY 10029, USA. [10]Present address: National Center for Immunization and Respiratory Diseases, Centers for Disease Control and Prevention, Atlanta, GA, USA. [11]Present address: Department of Microbiology and Molecular Medicine, University of Geneva, Geneva, Switzerland. [12]These authors contributed equally: Drishti Kaul, Raveen Rathnasinghe. ✉email: cdupont@jcvi.org; rmedinai@uc.cl

nfluenza A virus (IAV) is a highly infectious upper respiratory tract (URT) disease in humans and animals caused by a negative-sense segmented RNA virus. It is recognized as a major public health concern resulting yearly in significant disease and economic burden. Frequent nucleotide substitutions lead to changes on the hemagglutinin and neuraminidase glycoproteins on the surface of IAV particles (also known as antigenic drift) that contribute to the need for continuous vaccine updates. This evolutionary arms race between vaccine design and viral mutation contributes to annual influenza epidemics worldwide, which on average results in 3–5 million cases of severe illness and up to 291,000 to 646,000 deaths annually[1]. The modular architecture of the segmented IAV genome allows for genetic re-assortment (antigenic shift) with other divergent IAVs, resulting in the sporadic emergence of novel viruses capable of causing large epidemics or pandemics. Circulation of a new IAV in the naive human population has caused pandemics in the past resulting in significant morbidity and mortality, the most notable in 1918 and 1919, when the Spanish flu killed ~20 to 50 million people worldwide[2]. Retrospective analyses of autopsy specimens from the 1918 pandemic revealed the prevalence of secondary super-infection caused by URT bacteria[3–5]. However, the role of bacterial co-infection in disease prognosis is not only confined to pandemics; bacterial and virus co-infection during seasonal influenza epidemics are commonly associated with increase hospital admissions, severe disease, and deaths[6,7].

Although the microbiome of non-diseased individuals is relatively stable, IAV infection has been shown to increase the diversity of bacterial taxa that are present in the URT[8]. Specifically, IAV can cause changes in the relative abundances of *Staphylococcus* and *Bacteroides* genera[9], as well as *Haemophilus*, *Fusobacteria*, and other taxa[10]. Temporary disturbances to the microbiome due to the changes in the local epithelia during acute or chronic conditions has also been reported as a predisposing factor for infections[11–14]. The observed diversity in the human URT microbiome, together with its role in immunity and susceptibility to pathogens has been described previously[11,15,16]. Other studies have reported that the URT microbiome may also play a beneficial role in modulating the inflammatory response induced during IAV infection[16,17]. In addition, the intestinal microbiome composition has been shown to positively regulate the toll-like receptor 7 signaling pathway following infection with IAV[18]. Nonetheless, the exact mechanisms by which prior infection with IAV increase susceptibility to a secondary bacterial infection have not been determined. Importantly, the effect of IAV replication and induction of innate immune response on the composition of the human or animal URT microbiome remains to be elucidated and analyzed in depth on a community wide scale.

Humans and ferrets share similar lung physiology, and both are known to be susceptible and transmit the same strains of the IAVs[19,20]. This has made the ferrets an ideal model to study the dynamics of IAV infection in URT. However, it is unknown whether there is similarity between the ferret and human URT microbiome in terms of composition and its temporal dynamics and modulation upon IAV infection. In this study, we examine the longitudinal diversity of the URT microbiome of influenza-infected and uninfected human cohorts, as well as control uninfected and experimentally infected ferrets. These experiments reveal a strong consistency in the microbiome composition and dynamics between the two host systems, demonstrating that experimentally infected ferrets recapitulate closely the modulation of the microbiome observed in naturally infected humans. Our results suggest that microbiome disturbance and resilience dynamics may be critical to addressing the bacterial co-infections associated with influenza-derived morbidity.

## Results

**Effects of influenza on the human URT microbiome dynamics.** In order to determine if the human microbiome structure is modulated by the IAV infection, we established a human cohort study and obtained nasopharyngeal swabs at multiple time points after the initial influenza-prompted hospital visits (days 1–37 after initial onset of symptoms) from 28 human subjects recruited during 2011 and 2012. As healthy controls, we included nasal swab samples taken at six time points (days 1, 2, 3, 5, 7, and 28) from 22 healthy human subjects free of any respiratory infections (Supplementary Table 1). At the extreme age ranges, the case and controls were not age balanced, with a lack of healthy controls for the nine cases older than 65 years in age and the two cases that were younger than 5 years in age. Our goal was to assess and compare the temporal microbiome biodiversity in response to ecological disturbances of the URT caused by viral infection.

The dynamics and relative abundances of bacteria in the URT microbiome were examined by pyrosequencing of the V1–V3 region of the 16S rRNA, which yielded a total of 2.3 million sequences, which clustered into 707 operational taxonomic units (OTUs) (Table 1). The count abundance data for the OTUs was normalized to account for the sampling process and the library size, as confounding factors for the beta-diversity analyses. In addition, OTUs with counts <5 were removed to avoid inflating the importance of any contaminant sequences that might be present in the data. This resulted in over 90% of the reads mapped back to the OTUs (Table 1). Metric multidimensional scaling of the beta-diversity explains 38.5% of the variability across the first three components (Fig. 1). The plot shows that the IAV infection status has a strong influence on the ordination of the samples, as measured by the Bray–Curtis metric (ANOSIM $R = 0.696$, $p$-value < 0.001). The uninfected and infected communities cluster away from each other (Fig. 1). Of interest, the microbiome for the IAV-infected cohort is more dynamic than that of the uninfected IAV-free cohort, validating the Anna Karenina principle of microbiomes[21], which refers to the notion that there is much more variability in the microbial communities of infected (dysbiotic) individuals than in healthy individuals. The nasopharyngeal samples from infected humans demonstrated higher diversity between infection states than within them (Supplementary Fig. 1). The t-statistic for the "All within infection" versus "All between infection" for the human data set was −144.78, and the p-value was also significant (Supplementary Table 2), which indicates that IAV infection in humans results in the clustering of microbiomes according to infection status.

**Human URT dysbiosis is independent of clinical factors.** To complement the qualitative overview of the IAV-infected data

### Table 1 Summary statistics for amplicon-based sequencing of the V1–V3 region of the 16S rRNA gene.

|  | Humans | Ferrets |
|---|---|---|
| Total no. of samples[a] | 262 | 86 |
| Influenza-negative subjects | 22 | 7 |
| Influenza-positive subjects | 28 | 7 |
| Total no. of reads | 2,300,072 | 649,440 |
| Total no. of OTUs | 707 | 259 |
| No. of reads mapped to OTUs | 2,111, 66 (91.8%) | 514,099 (79.2%) |

[a]All human and ferret samples were extracted from nasal washes and nasopharyngeal swabs, respectively, at several time points post symptom onset (humans) or post infection (ferret).

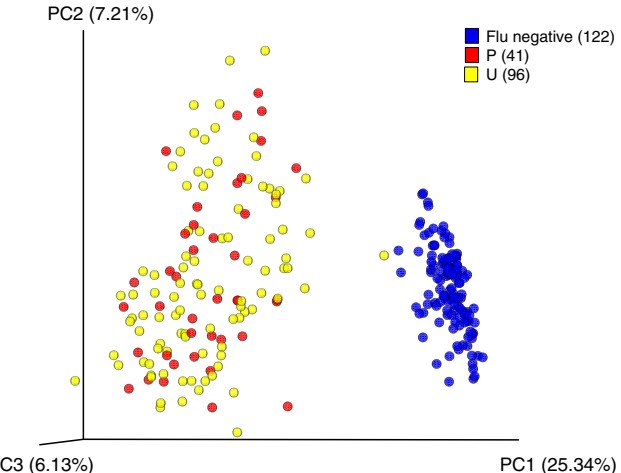

PC2 (7.21%)

PC3 (6.13%)    PC1 (25.34%)

Flu negative (122)
P (41)
U (96)

**Fig. 1 Diversity of the URT microbiome of human patients infected with influenza A virus (IAV).** Beta-diversity analysis for longitudinal nasopharyngeal swab samples obtained from healthy and IAV-infected individuals. Principal coordinates analysis (PCoA) of Bray–Curtis distances was done for samples from humans, labeled as influenza positive in red (P, indicating data points with positive IAV qRT-PCR detection), influenza unknown in yellow (U, indicates time points from positive individuals that were below the qRT-PCR detection limits at different time points after the onset of symptoms), and uninfected samples in blue (Flu negative). The total variability explained by all three principal coordinates (PCs) is shown on the axes. Source data are provided as a Source Data file.

points, we integrated additional clinical metadata including gender, antibiotic usage, and age, and included details of the amplification of IAV genomes from these samples to more accurately classify these data points as either positive or unknown for the presence of virus. Positive and unknown infected microbiomes were tested to determine if they were distinct enough to cluster separately based on their beta-diversity. Analyses of the beta-diversity metrics using PCoA, focusing just on the IAV-infected samples, did not allow deriving any conclusions from this analysis alone. In addition, the grouping of infected samples based on gender did not show any significant association (ANOSIM $R = 0.01$, p-value < 0.118 d$f$ 1), implying that there was no significant effect of gender on the clustering of the samples (Supplementary Table 3). When we used distances between the samples as the response variable (ADONIS d$f$ 1, $R^2 = 0.009$), only 0.9% of the variation in the distances was explained when the gender of the patients was accounted for as a predictor of the model. Hence, sex could not be correlated with the microbiome of the infected human samples. Age and effects of post visit antibiotic treatment on the microbiome trends were also examined. Little association could be observed between post visit antibiotic usage and clustering of the infected human samples in two statistical tests (ANOSIM d$f$ 1 $R = 0.242$, p-value < 0.001, and ADONIS d$f$ 1, $R^2 = 0.042$), which was surprising. However, the age of the patients seemed to have some influence on the sample grouping when all 26 categorical values were taken into consideration (ANOSIM d$f$ 37 $R = 0.402$, p-value < 0.001). The statistical analyses show that while the p-value was significant, the clustering on the basis of age was only moderately strong (ADONIS $R^2 = 0.427$, d$f$ 37; Supplementary Table 3). Since there was no indication of this effect among IAV-infected patients in the ordination plots (Supplementary Fig. 2), it is possible that the significant p-value could be attributed to the high number of samples or the differences in dispersion among the different sample groupings, emphasizing the importance of considering in the analysis both the p-value and the effect size. Finally, while

both vaccination status (ADONIS d$f$ 2 $R^2 = 0.24$, p-value < 0.001) and viral subtype (ADONIS d$f$ 2 $R^2 = 0.25$, p-value < 0.001) were examined and found to be significant, there was little indication of a real grouping in the ordination plot (Supplementary Fig. 2).

**_Pseudomonas_ blooms during viral infection in the human URT.** We examined taxonomic profiles for all the infected and healthy patients across all the time points using the taxa abundance values for the top ten most prevalent taxa at the class level, sorted by the most prevalent taxa in each cohort; _Gammaproteobacteria_ in the infected patient cohort and _Actinobacteria_ in the healthy patient cohort (Fig. 2). All other taxa were pooled into an additional taxon named "Other". _Pseudomonas_ was the most abundant taxonomic group in all samples from influenza-infected individuals (Fig. 2; and Supplementary Figs. 3 and 4). A phylogenetic inference places this OTU robustly as the genus _Pseudomonas_; however, this analysis cannot resolve the OTU to the species level (Supplementary Fig. 5). Less-abundant phyla included _Bacteroidetes_, _Firmicutes_, _Actinobacteria_, and some other families of _Proteobacteria_, like _Rhodanobactereceae_ and _Pasteurellaceae_ (c. _Gammaproteobacteria_) and _Brucellaceae_ of the _Rhizobiales_ order (c. _Alphaproteobacteria_). _Pseudomonas_ was also clearly identified as the predominant taxon when temporal dynamic analyses were done on individuals independently (Supplementary Fig. 6). As for the uninfected subjects, _Actinobacteria_ was the most dominant taxon, and _Pseudomonas_ was the least-abundant taxonomic group present, also seen when individual subjects were analyzed (Supplementary Fig. 6). Other less-abundant phyla included _Verrucomicrobia_ and within the _Proteobacteria_, the _Alphaproteobacteria_, and _Epsilonproteobacteria_ classes.

**IAV infection groups the human URT microbiome into ecostates.** Due to the dynamic nature of the human URT microbiome during IAV infection, we hypothesized that infection perturbs the microbiome structure resulting in distinct signature microbiomes that differentiate infected from uninfected individuals. Thus, we used the Infinite Dirichlet-multinomial Mixture Model (iDMM)[22], which is an extension of the Dirichlet-multinomial mixture model (DMM)[23] that helps understand and interpret taxon abundance data by adding statistical validation if a taxa is associated with a given case–control condition. This is an unsupervised clustering method that applies Bayesian statistics to quantitatively assess the data and accurately capture the features that are present. Essentially, given a set of subsampled distributions, the iDMM model predicts the original number of full-size distributions together with their composition. The nonparametric nature of the iDMM model makes it ideal for understanding the complex ecological data in this study, where the original number of the sampled communities (known as ecostates) is unknown.

The iDMM model was run over 2000 iterations over all data points (50 patients at multiple time points), which collapsed the data into a total of four ecostates (Table 2). Plotting the mean of the likelihood ratio at each iteration showed that, 25 iterations into the analysis, the maximum likelihood ratio converges for the model. One of the four ecostates included all 127 uninfected data points (or the healthy ecostate), while the 135 infected data points were distributed across the three other ecostates (or unhealthy ecostates). Interestingly, a few patients moved from the unhealthy ecostates during acute influenza infection to the healthy ecostate in the later time points. This suggests that the human microbiome exhibits resilience but potentially a weak elasticity; however, this could be due to the lack of a precise temporal control of the time of infection.

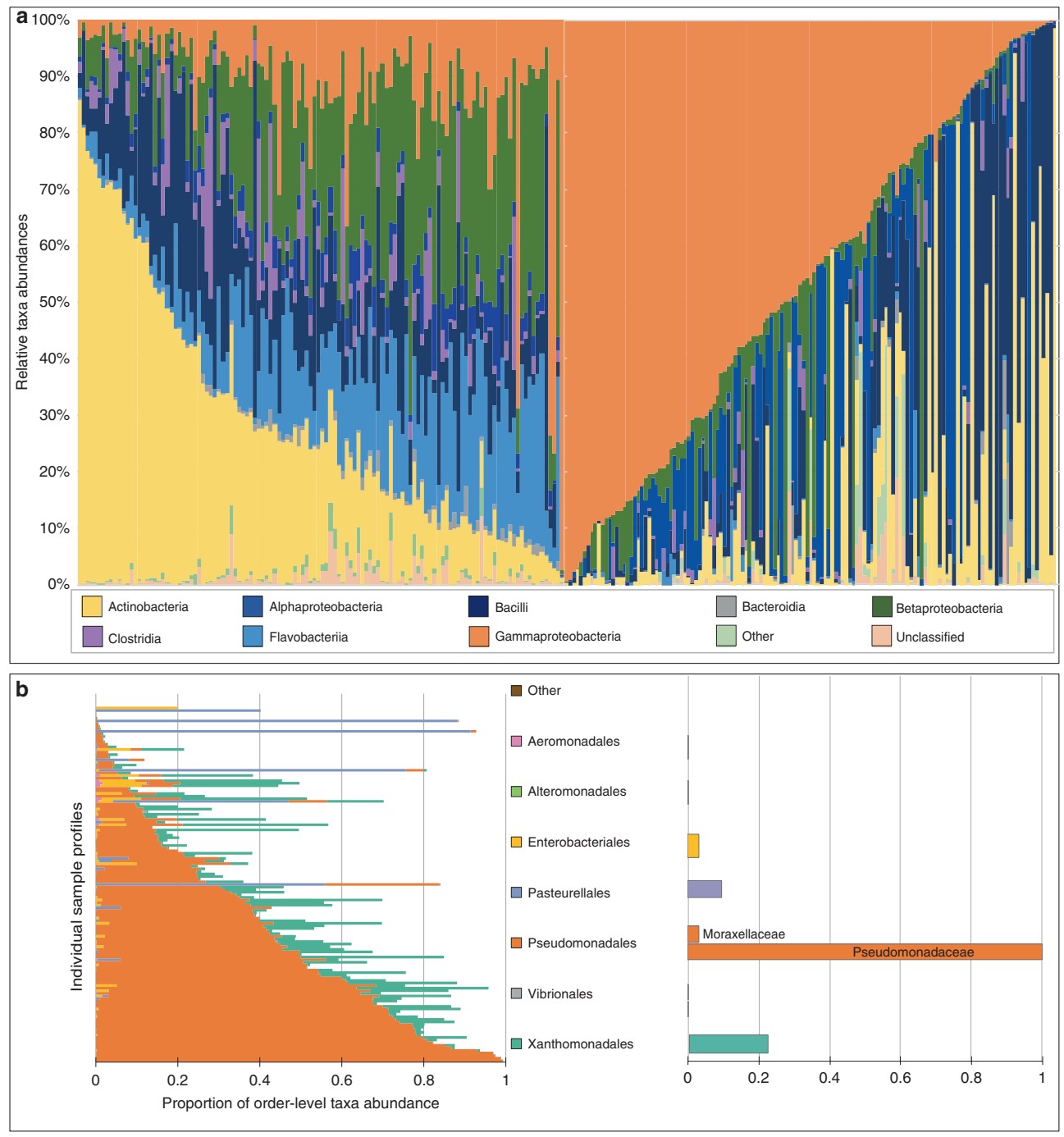

**Fig. 2 Comprehensive taxonomic breakdown for IAV-free (left) and IAV-infected (right) human subjects. a** Each column is a specific sample. The plot summarizes the relative taxonomic abundances at the class level for taxonomic groups that are present in >5% of the samples (see legend below), sorted in decreasing order by the most abundant taxonomic group in each cohort. Gammaproteobacteria (*Pseudomonas*, orange) bloom is prevalent among the infected patients (right), whereas Actinobacteria is the most abundant among healthy patients. **b** An order (left) and family (right) level breakdown of the Gammaproteobacteria observed in infected patients. Source data are provided as a Source Data file.

We also identified a diagnostic OTU for each of these ecostates, which is the OTU with the highest posterior-predictive probability in the ecostate and therefore drives the clustering. The iDMM analysis predicted the diagnostic OTU for the healthy ecostate to be Otu000008 which belongs to the *Flavobacteria* class (*Cloacibacterium*), with a posterior-predictive probability of 0.08, followed by Otu000010 (*Corynebacterium_1*) and Otu000013 (*Comamonadaceae*), belonging to the class Actinobacteria and Betaproteobacteria, respectively (Table 2). For the unhealthy

ecostates, Otu000003, Otu000004, and Otu000002 were diagnostic for Ecostate 1, 2, and 3, respectively (Table 2). Ecostate 1 had the largest number of infected data points (104), followed by Ecostate 3 (20) and Ecostate 2 (8). Otu000003 and Otu000002 belong to the *Pseudomonadaceae* family (the latter being an unclassified *Pseudomonadaceae*), with relatively high posterior probabilities associated with each of them (Table 2). Otu000004 belonged to the *Actinobacteria* class, and was the diagnostic OTU for Ecostate 2 with eight infected data points. The diagnostic

**Table 2 Diagnostic microbes for each ecostate from the 2000th iteration of the iDMM model for the infected and uninfected humans.**

| Ecostate | Final distribution[a] | Original sample distribution[b] | Diagnostic OTU | Probability associated[c] | Taxonomy |
|---|---|---|---|---|---|
| 1 + 2 + 3 (infected) | 104 | 146 | Otu000003 | 0.361568 | Bacteria; Proteobacteria; Gammaproteobacteria; Pseudomonadales; Pseudomonadaceae; Pseudomonas |
| | 8 | | Otu000004 | 0.4989514 | Bacteria; Actinobacteria; Actinobacteria; Corynebacteriales; Corynebacteriaceae; Corynebacterium_1 |
| | 20 | | Otu000002 | 0.01584407 | Bacteria; Proteobacteria; Gammaproteobacteria; Pseudomonadales; unclassified |
| 4 (healthy) | 130 | 127 | Otu000008 | 0.07636954 | Bacteria; Bacteroidetes; Flavobacteriia; Flavobacteriales; Flavobacteriaceae; Cloacibacterium |

[a]Distribution of samples within ecostates after running the iDMM model.
[b]Distribution of samples before running the iDMM model.
[c]Bayesian posterior-predictive probabilities associated with the diagnostic microbe, which is the highest probability for that ecostate.
The number of iterations depends on the number of samples (273) present in the data. Source data are provided as a Source Data file.

OTUs for all four ecostates for the human samples are also among the first ten most abundant OTUs for the data.

A random-forest analysis was also used to identify predictive features in the data. The method we developed iterates through unique random-forest models (each seeded with a different random state), and attempts to fit the model to a random subset of the data with five samples removed from the training set (see "Methods"). If the model could accurately predict all five of the omitted samples during the cross-validation step, then its feature importance vector (mean decrease gini index), including weights for every OTU's predictive capacity, was collected. The results from the random-forest classification aligned with our diagnostic iDMM OTU prediction in the human samples (Supplementary Table 4). The analysis showed Otu000002 (unclassified *Pseudomonadales*) to be the most predictive of the IAV-infected samples, followed by Otu000001 (*Rhizobiales*) and Otu000003 (*Pseudomonas*) with a maximum accuracy of 71%. When we examined the taxonomy of Otu000001 in detail, it was classified with 100% confidence down to Genus *Ochrobactrum*, at which point the read length is unable to differentiate the species any further. Nevertheless, the actual OTU sequence is 100% identical to *Ochrobactum anthropi*, an opportunistic human pathogen[24–26]. Similarly, the in depth analyzes of Otu000006 identified the taxonomy of this OTU as uncultivated lineages of Rhodanobacter, which have also been previously associated with human respiratory tract microbiomes[27]. Comparison with our negative controls confirmed that these were not contaminants and supported the notion that *Ochrobactrum* was also diagnostic for the infection state in humans, which is likely to be consistent with the presence of *O. anthropii* or similar opportunistic species.

**IAV infection modulates the ferret URT microbiome structure.** We hypothesized that IAV infection in ferrets will result in the clustering of microbiomes according to infection status, as observed during IAV infection in humans. Therefore, using the well-established ferret model of IAV infection, we designed a longitudinal study resembling the clinical specimens obtained from human patients to obtain nasal wash samples from infected animals. We collected nasal washes from seven uninfected ferrets and seven ferrets infected with the A/Netherlands/602/2009 (H1N1) pandemic strain, at 0, 1, 3, 5, 7, and 14 days post infection (dpi). The dynamics and relative abundances of bacteria in the URT microbiome were examined by pyrosequencing of the V1–V3 region of the 16S rRNA using similar thresholds for length, and expected error as was chosen for the human data. A

total of 649,440 reads clustered into 259 (OTUs) with 79% of reads mapping (Table 1). As before, the count abundance data for the OTUs was normalized, and the low abundance taxa were filtered out from the count data. Principal coordinates analysis (PCoA) of beta-diversity between the healthy and IAV-infected groups demonstrated variability consistent with the virus perturbing and modulating the microbiome structure (Fig. 3). Infection status strongly influenced the ordination of the samples as measured by the Bray–Curtis beta-diversity metric ($R = 0.503$, $p$-value $< 0.001$). The IAV-negative and IAV-positive ferret microbial communities formed discrete clusters, while samples from the IAV-infected animals showed divergence from each other (Fig. 3). By the final time point, day 14, the microbiome of infected ferrets (light blue) was more similar to the day 0 samples (lavender) and those of the uninfected controls (dark blue).

Quantitative metrics of diversity were used to compare the microbiomes of influenza-infected and control ferrets. Beta-diversity distance analyses (Supplementary Fig. 7) demonstrated that ferret microbiomes had higher diversity between infection states than within them. Student's two sample two-sided $t$ tests confirmed that the diversity between the two states (infected and uninfected) was statistically significant, with the microbiomes of infected ferrets being more diverse (Supplementary Table 5). The t-statistic for the "All within infection" versus "All between infection" was $-28.681$ corresponding to a Bonferroni-corrected parametric $p$-value of 8.85e-161 (Supplementary Table 5). The PCoA and statistical analyses showed that infected ferrets have a far more dynamic URT microbiome than that of the uninfected group. A separate healthy baseline experiment was conducted, in which we identified some divergence of the microbiomes in the absence of infection, hence differences in the microbiome structure of each animal was expected given the high level of personalization, and that ferrets are outbred. Remarkably, 7/7 $T = 14$ time points converged to the healthy microbiome, together with 4/7 $T = 0$ time point samples. Overall, the quantitative examination revealed that the range for infection-associated beta-diversity was much lower in the ferret samples than it was from human clinical samples.

**IAV induces temporal changes in the ferret URT microbiome.** To assess the correlation of clinical symptoms over time during acute IAV infection, we monitored the body weight of all ferrets from 0 to 14 dpi, which demonstrated a clear weight loss among the infected animals (Fig. 4a). As expected, the maximum weight loss coincided with peak IAV titer from 3 to 5 dpi, and recovery in body weight correlated with the lack of detectable virus after

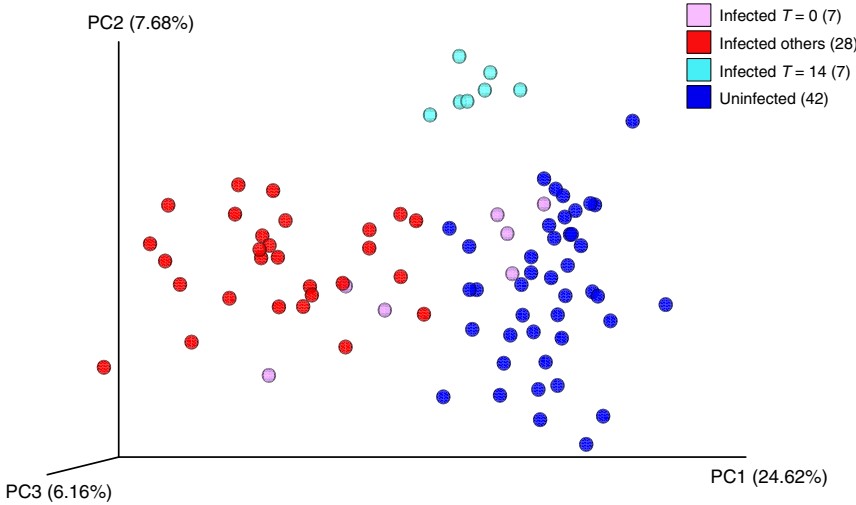

**Fig. 3 Diversity of the URT microbiome in ferrets during IAV infection.** Beta-diversity analysis for longitudinal URT samples taken after experimental infection with the A/Netherlands/602/09 H1N1 strain (Infected) or in control animals. Principal coordinates analysis (PCoA) of Bray–Curtis distances was performed for all samples. Data points for uninfected ferrets are in blue, the $T = 0$ for the infected ferrets in lavender, the $T = 14$ for infected ferrets in cyan, and all other infected time points were collected at 1–7 days post infection and shown are in red. The total variability explained by all three principal coordinates (PCs) is shown on the axes. Each group of ferret was composed of seven animals. Source data are provided as a Source Data file.

day 7 (Fig. 4b). To better visualize the temporal trajectory of the ferret microbiome, the community composition for two influenza-infected (with divergent baseline microbiomes) and one uninfected ferret (ferret_595 and ferret_587, and ferret_592, respectively) were examined with regards to their taxonomic profiles across six different time points (Fig. 4c–e). At the order level, the IAV-infected ferrets exhibited peak *Pseudomonadales* abundance at days 5 and 7 dpi (Fig. 4c–g), which correlated with maximal weight loss and peak viral titers (Fig. 4a, b), suggesting the direct or indirect influence of the infection on the microbiome. A phylogenetic inference shows this OTU to be in the order *Pseudomonadales*, but belonging to the genus *Acinetobacter* (Supplementary Fig. 5). A few of the less-abundant phyla included *Actinobacteria* and *Firmicutes* (Supplementary Fig. 8). The abundance of *Pseudomonodales* decreased over time in the infected ferrets, reaching the basal abundance found in healthy ferrets 14 dpi. For the uninfected ferrets, the microbiomes were more stable and *Clostridiales* was the most abundant taxonomic group, followed by *Lactobacillales* (light blue). *Pseudomonadales* were among the least-abundant taxonomic group in the uninfected controls (Fig. 4e). This was also observed when we analyzed the microbiome abundance of each individual animal in both infected and uninfected groups (Supplementary Fig. 9). These results demonstrate that IAV infection induces a dynamic modulation of the microbiome structure in the URT of ferrets, which correlated with viral replication and pathogenesis.

**IAV infection groups the ferret URT microbiome into ecostates.** Since the timing of infection was controlled in the ferret experiment, we hypothesized that upon infection the microbiome structure would be ordered into more defined ecostates for the infected and uninfected animals. Hence, we run the iDMM model over 1000 iterations, which collapsed the data into two ecostates. The mean of the likelihood ratio at each iteration converged 70 iterations into the analysis, splitting into two ecostates until the last iteration. Of interest, one of the two ecostates comprised all the uninfected data points (or the healthy ecostate), while the other contained most of the influenza-infected data points (the unhealthy ecostate, Table 3). There were notable exceptions; despite the perturbation caused by the infection, all day 14 samples in the infected cohort moved from the unhealthy ecostate to

the healthy ecostate, which is also shown in the ordination plot (Fig. 3). The healthy ecostate also contained a few of the earlier data points (day 0 and day 1) of the influenza-infected cohort, indicating a temporal lag in changes to the ferret microbiome at those time points when the IAV titer was submaximal (Fig. 4b).

The iDMM analysis for ferrets predicted the diagnostic OTU for the unhealthy ecostate to be Otu000004 that belonged to the *Pseudomonadales* order, with a posterior-predictive probability of 0.11 (Table 3), followed by Otu000003 with the next highest predictive probability of 0.08, belonging more specifically to the *Pseudomonas* genus (Supplementary Fig. 8). This is consistent with the qualitative taxonomic profiling (Fig. 4). For the healthy ecostate, Otu000001, which belongs to the *Clostridia* family, was the diagnostic OTU with a posterior-predictive probability of 0.19 (Table 3). The posterior probabilities for each taxon were calculated within each sample by observing the fraction of simulated samples with more counts than the observed value. The probabilities associated with the diagnostic OTUs can be thought in terms of being relative to all taxa present. Similar to the human data, the diagnostic OTUs for both ecostates are among the ten most abundant OTUs for the data (Supplementary Fig. 8). This was also confirmed when the microbiome for all ferrets from both infected and uninfected groups was analyzed individually (Supplementary Fig. 9), which indicates that *Pseudomonadales* are not only predictive of the unhealthy ecostate but also undergo the greatest temporal dynamic change during IAV infection. This was confirmed when alpha-diversity analyses were conducted, which showed a drastic decrease in diversity by day 7 (Supplementary Fig. 10). The results from the random-forest analysis aligned well with the iDMM diagnostic OTU prediction in that Otu000004 (*Pseudomonadales*) was the most predictive attribute for the samples from IAV-infected ferrets, followed by Otu000028 (*Enterobacteriaceae*) and Otu000017 (*Bacillales*), with a maximum accuracy of 96% (Supplementary Table 6). Altogether, these data indicate that IAV infection results in a nasal bloom of multiple *Pseudomonadales* in the ferrets, displacing the *Clostridia* associated with the healthy and stable ecostate.

**Discussion**

This longitudinal study describes taxonomic microbiome population dynamics in the upper respiratory tract of humans and

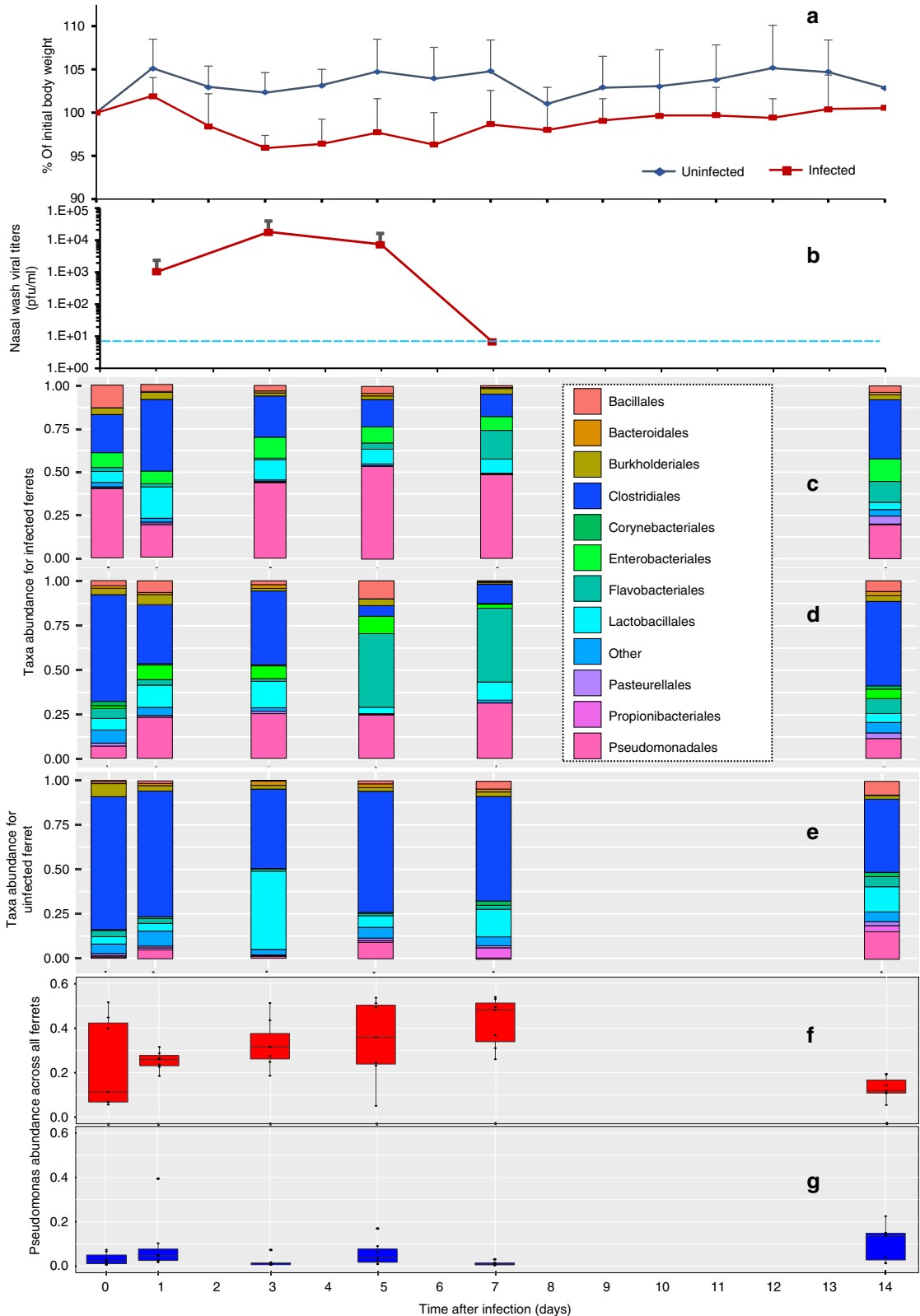

ferrets during IAV infection. Given the unequivocal association between viral and bacterial co-infection and influenza disease severity, there is a pressing need to better understand how perturbation of the host microbiome correlates with viral infections that facilitate opportunistic co-infections. The nature of the 16S sequencing approach taken, that is a loci-based population survey, means that we can address taxonomy-centric ecological principles of disturbance and resilience[28,29] in the URT microbiome. Our results strongly suggest that the core URT microbiome is perturbed by IAV infection via direct and uncharacterized indirect processes, which may in turn might facilitate co-infections with bacterial pathogens causing increased

**Fig. 4 Qualitative and quantitative representation of the temporal trajectory of the ferret microbiome. a** Percent body weights of groups of seven ferrets mock inoculated (uninfected) or intranasally infected with $1 \times 10^6$ pfu of influenza A/Neth/602/09 virus. Body weights were determined daily for 14 days, and are represented as the average percent body weight compared with the initial weight of each animal on the day of inoculation and error bars are the standard deviation for each time point. **b** Viral titers of nasal washes of ferrets infected with $1 \times 10^6$ pfu of A/Neth/602/09 virus. Nasal washes were obtained on days 1, 3, 5, and 7 post infection, and are represented as the average viral titer of seven infected animals. Error bars indicate the standard deviation for each time point. The limit of detection of the assay was 10 pfu/ml (dashed line). **b**–**d** Comprehensive taxonomic breakdown of two influenza-infected, both representing divergent baseline microbiomes (**c**, **d**) and uninfected ferret (**e**), at different time points. Taxa abundance values for top ten most prevalent taxa at the order level for different time points (0–14 dpi). Only taxa labels with a confidence score of > = 90% were retained in the analysis. The remaining taxa are pooled into an additional taxon labeled "Other". **f**, **g** The boxplots represent the relative *Pseudomonas* abundance across all infected (**f**) and uninfected (**g**) ferrets (*n* = 7 for each). The box represents the interquartile range, the horizontal line within the box indicates the median for each sample grouping, observations are indicated by dots, and the whiskers outside the box extend from the highest to the lowest observation represented in the plot. Source data are provided as a Source Data file.

| Table 3 Diagnostic microbes for each ecostate from the 1000th iteration of the iDMM model for the ferret samples. | | | | | | |
|---|---|---|---|---|---|---|
| Ecostate | Total samples | No. of samples[a] T14 [T7 + T5 + T3 + T1] T0 | | Diagnostic OTU | Probability associated[b] | Taxonomy |
| 1 (healthy) | **58** (42) | **14** (7) **33** (28) **11** (7) | | Otu000001 | 0.1865749 | *Bacteria; Firmicutes; Clostridia; Clostridiales; Peptostreptococcaceae; Romboutsia* |
| 2 (infected) | **26** (42) | **0** (7) **23** (28) **3** (7) | | Otu000004 | 0.1112045 | *Bacteria; Proteobacteria; Gammaproteobacteria; Pseudomonadales; Moraxellaceae; Acinetobacter* |

[a]The number of samples at final iteration for each time point in bold (original starting values in parentheses).
[b]Bayesian posterior-predictive probabilities associated with the microbe, which is the highest probability for that ecostate.
The number of iterations depends on the total number of samples (84) present in the data. All later time point ferrets (T14) return to the healthy ecostate (1). Source data are provided as a Source Data file.

hospitalizations and morbidity associated with IAV infection. In addition, our results provide a clear approach for the design of future studies explicitly examining the mechanistic links between IAV and bacterial co-infection, along with the development of therapeutic treatments aimed at the microbiome as a community.

Without disturbance or perturbation, the URT microbiome was stable in both uninfected humans and ferrets. IAV does not directly infect any microbiome constituents, yet infection disturbs the healthy-state microbiome in both hosts in a statistically robust manner. The microbiomes of infected (unhealthy) individuals or animals were quite different from each other (Figs. 1 and 3). However, in both hosts, unhealthy microbiomes were divergent from the healthy microbiomes, and numerous community assemblies were possible in the unhealthy state. This is a clear demonstration of the Anna Karenina principle[21], restated as "all healthy microbiomes are the same, while unhealthy microbiomes are unique." This high diversity of unhealthy microbiomes during early stages of acute infection is consistent with earlier studies[8], but here we demonstrate specifically that it can occur as a consequence of an indirect disturbance such as IAV infection. In agreement, in recent studies changes in the microbiome structure were also reported in a household contact setting in secondary cases of IAV infection[30,31], and in a cohort of infected individuals with either H3N2 Influenza Infection or individuals infected with Influenza B[32]. We propose that the disturbance of the healthy URT microbiome creates transient ecological niches for opportunistic bacterial pathogens. How viral infection induces a disturbance in the microbiome requires further assessment. Nevertheless, the host antiviral responses such as the induction of interferon during IAV infection could contribute to the perturbation of the microbiome in a dynamic manner, though this requires host and microbiome metatranscriptomics or metaproteomics measurements in controlled experiments focused at the onset of infection. Nevertheless, maximum disturbance correlated with maximum viral loads and weight loss in the ferret model, which suggests a close relationship between active infection,

disease, and disturbance of the microbiome, with kinetics that are similar to the antiviral response induced during IAV infection[33]. This is in contrast to a recent study of previously healthy individuals experimentally infected with a H3N2 influenza strain from 2005, where no oropharyngeal microbiome changes were reported[34]. This discrepancy might be due to the fundamentally different nature of the oro- and nasopharyngeal microbiomes[35], to potential differences induced by natural versus experimental infection in humans, or due to intrinsic differences induced by different IAV subtypes. No significant influence of the host type (age and sex) or behavior (antibiotic usage) was observed on the temporal nature of the microbiome elasticity. However, more statistical power would be needed to draw any further robust associations from the data. This is particularly the case for our cases under 5 years of age and greater than 65 years of age, for which we lacked healthy age-matched controls. Hence, additional human cohort studies to underpin the factors modulating microbiome dynamics during IAV infection are warranted.

The sole statistical exception to the high community diversity of infected microbiomes was the increased relative abundance of *Pseudomonadales*, regardless of age, sex, antibiotic treatment, or even host organism. Yet, the bloom of *Pseudomonadales* is consistent with previous reports in H1N1-infected patients[9,15,36,37]. It must be noted that, while the DNA extraction method used could result in a slight underrepresentation of Gram-positive bacteria, this bias would be consistent across both infected and healthy control samples. In our study, *Pseudomonadales* are present in relatively low proportions in the healthy microbiome of these host organisms. Therefore, their bloom might be due to a more hostile environment for the other taxa or perhaps a more hospitable environment for the *Pseudomonadales*, making this an excellent candidate for future strain isolation, genome sequencing, and transcriptional profiling. The differences in relative abundance observed between the infected cohorts opens the idea of using treatments capable of modulating the microbiome back into the healthy ecostate[38]. Such a treatment would be

homologous to those proposed for perturbing or restoring the gut microbiome[39]. Understanding how and why *Pseudomonadales* succeed after disturbance will provide valuable information for conducting future microbiome centric URT studies in a controlled setting. It should be noted that the blooming *Pseudomonads* are not *P. aeruginosa* (Supplementary Fig. 5), and understanding their functional potential and role requires shotgun metagenomics analyses for more detailed phylogenetic and functional profiling.

In addition, in humans secondary *Pseudomonas* infections have been extensively described before, and *Pseudomonas* infections have been specifically linked to nosocomial infections as a result respiratory support treatments in hospital settings[40–44]. It is currently unknown whether infection with other respiratory viruses can also induce the modulation of the URT microbiome; however, since severe viral infections often require respiratory support, including intubation, it is likely that co-infection with pathogens such as the *Pseudomonadales* could actually be favored due to previous perturbations of the microbiome. Hence, additional associative studies to elucidate factors that modulate the temporal change of the microbiome structure could also aid in understanding the factors that promote or support secondary bacterial colonization during severe respiratory viral infections.

In the ferret model, there is a clear demonstration of ecological resilience in the URT microbiome; namely a return to the original community after disturbance, a phenomenon also observed, albeit less clearly, in the human samples, which had an unknown and likely more diverse ecostate prior to infection. Similar observations have been reported in the human gut microbiome after the disturbance associated with antibiotic treatment[28], though our findings expands it to the URT and the indirect effects of the IAV infection. The controlled experiments with ferrets resulted in near complete recovery. Human URT microbiomes do not unequivocally show a return to the health state, but in several patients, the microbiome returned to the healthy ecostate. Although it is tempting to suggest that the ferret microbiome might have greater elasticity (i.e., less time required for demonstration of resilience), there are multiple potential reasons for the discrepancy between ferrets and humans. For instance, the routes of and mode of infection are different, with the ferrets receiving a high volume containing a high titer dose intranasally, whereas human likely get infected by the aerosol route at potentially lower titer. On the other hand, considering metabolic rate relative to organism size, the ferret may recover at a more rapid rate simply due to a higher metabolism. More pertinently, the human cohort has an undetermined infection date, were infected by different viral strains (and viral variants as determined by whole IAV genome sequences) and had a selection bias towards phenotypically responsive patients (e.g., symptomatic hospitalized patients), where zero time (day 0) was the first day of symptom. Beyond the potential differences in absolute temporal trends in microbiome resilience and elasticity, the human and ferret microbiomes share similar trends at the ecosystem and individual taxon level that warrant further experimentation. The results here provide an experimental baseline for examining both predictive and therapeutic intervention focused experiments in the ferret model system. For example, the presented hypothesis that IAV driven microbiome disturbance increases the propensity for bacterial pathogen co-infection can be robustly tested by bi-partite exposures to viral, and then bacterial pathogens. The effects of lifestyle (diet, smoking, exercise) and abiotic influences (humidity, temperature) on the microbiome and its resilience should also be examined, particularly with regards to temporal dynamics of microbiome disturbance and recovery. Potential therapeutic

approaches involve thwarting the associated threat of opportunistic bacterial pathogens or interventions focused on the bloom of *Pseudomonas*, where probiotic treatments could be explored to maintain the homeostasis as seen in the healthy individuals. Our results are especially relevant in the context of secondary bacterial infections following primary infection with IAV[45]. Multiple studies, including this one, have now shown that a subset of the taxa that are most frequently associated with secondary infections have increased relative abundance during IAV infection. It is possible that such outcomes could be reduced by modulating the host immune response during IAV infection[17]. Reducing the high morbidity and mortality rates associated with such secondary infections would improve quality of life and longevity while simultaneously reducing healthcare costs[40,46,47].

## Methods

**Human sample collection and study design**. Patient clinical–epidemiological data, along with nasopharyngeal swabs were collected after informed written consent was obtained under protocols 11-116 and 16-066, reviewed and approved by the Scientific Ethics Committee of the School of Medicine at Pontificia Universidad Catolica de Chile (PUC) before the start of sample collection. Between July and August of 2011 and June and September of 2012 (during the Southern Hemisphere autumn–winter season), a total of 146 nasopharyngeal swabs samples were collected from 30 hospitalized patients in Santiago, Chile, diagnosed clinically with influenza-like illness (ILI). Of the 30 patients in the study, 28 were confirmed and subtyped as H1N1pdm09 or H3N2 Influenza through RT-PCR by the Clinical Virology Laboratory at PUC. The remaining two patients could not be confirmed as influenza positive by qRT-PCR, RT-PCR and/or the hemagglutination inhibition (HI) assay, so they were not included in further analyses. Samples were also tested against 13 other pathogens. In only two cases, we detected co-infections at the day of recruitment, one patient had Rhinovirus and another had Respiratory Syncytial virus; however, they tested negative in subsequent test. Between one and six samples from the acute phase of infection were taken from each patient, together with a sample up to 22 days post diagnosis (convalescence phase or healthy baseline) from 14 out of the 28 individuals analyzed. Upon collection of all samples, the timing of the infected cohort samples was established as the time in days since the onset of symptoms. Control samples from 22 healthy individuals, confirmed as negative against influenza A virus and 13 other common respiratory viruses, were obtained at the outpatient clinic with the same criteria in March to June of 2014. Epidemiological history, signs and symptoms, other diagnostics and treatments of each patient were also collected during hospitalization as detailed in Supplementary Table 3. Furthermore, 96.4% of patients received oseltamivir antiviral treatment, and 89.3% received antibiotics originating from the families of the fluoroquinolones (levofloxacine, morifloxacine, or ciprofloxacine), 3rd-generation cephalosporins (ceftriaxone or cefepime), carbapenems (meropenem or imipenem), metrodinazole, cotrimoxazole, or vancomycin. These treatments where supplied in a combination of 5 (4% of patients), 4 (8%), 3 (12%), 2 (40%), or one (36%) antibiotics in a complete treatment (at least 7 days) or less. Severe infection criteria were established in accordance with the hospitalization of influenza and/or derivation to critical care unit (which involves oxygen support or mechanical ventilation and/or vasoactive drug administration) after symptoms onset. The microbiome data analyzed were obtained from the nasopharyngeal swabs of 28 infected subjects (13 male and 15 female), ages ranging from 1 year to 76 years, for a total of 121 samples. The naming convention of influenza A viruses detected from patients are as follows: A/Santiago/p*x*d*y*/2011 or A/Santiago/p*x*d*y*/2012 (p = patient and d = day). The negative controls analyzed in the study were nasopharyngeal swabs taken from 22 healthy patients (10 males and 12 females), ages ranging from 19 year to 65 years, most taken at all six time points (1, 2, 3, 5, 8, and 28 post enrollment), for a total of 127 samples, which were negative for influenza and other respiratory infections.

**Ferret infection and sample collection**. The animal experiments described here were performed under protocols approved by the Icahn School of Medicine at Mount Sinai Institutional Animal Care and Use Committee, adhering strictly to the NIH Guide for the Care and Use of Laboratory Animals. Influenza-free and specific pathogen-free 6-month-old female ferrets (*Mustela putorious furo*) were purchased from Triple F Farms. The animal's sera were confirmed to be negative against circulating H1N1, H3N2, and B influenza viruses before they were shipped from the company. Upon receipt, the animals were handled only by trained personnel wearing an N95 mask (that prevents the transmission of airborne pathogens), and were immediately housed individually in PlasLabs poultry incubators fitted with high-efficiency particulate air (HEPA) filters to provide them with pathogen-free air through out the experiment. Prior to the start of the experiment, the animals were allowed to acclimate for 48 h before nasal inoculation. They were also

provided with access to food and water ad libitum. All infections and nasal wash samples were done on ferrets anesthetized with ketamine (25 mg/kg) and xylazine (2 mg/kg) intramuscularly. A detailed time point study was conducted in ferrets infected with $1 \times 10^6$ plaque-forming units diluted in a final volume of 0.5 ml of sterile PBS per animal of the A/Netherlands/602/2009 H1N1 pandemic strain through intranasal inoculation. Control animals were mock infected only with 0.5 ml of sterile PBS. Then nasal wash samples were taken from the seven uninfected and seven infected animals. To study the effect of IAV infection on the URT microbiome, samples were taken at six different time points: on day 0 (1 h post inoculation) and then on days 1, 3, 5, 7, and 14 post infection (dpi). Body weights were obtained for 14 consecutive days, and viral titers were determined by plaque assay in MDCK cells as previously described[48] for the first 7 dpi.

**Sample processing, and sequence analyses**. All bacterial genomic DNA (gDNA) extractions were performed using the Qiagen All Prep kit, and were subjected to 16S amplification using the HMP 16S sequencing protocol, and the amplicons were sequenced using the Roche 454 Titanium pipeline[49]. Appropriate positive and negative controls from amplification were also included. The V1–V3 hypervariable regions were amplified for 16S profiling (forward primer: 27F 5′-AGAGTTT-GATCCTGGCTCAG-3′ and reverse primer: 534R 5′-ATTACCGCGGCTGCTGG-3′) of the 16S ribosomal RNA gene.

**Data analysis**. Reads were de-multiplexed according to barcodes followed by trimming of both barcodes and adapter sequences. Following the initial processing of the sequence data, sequences were combined, dereplicated, and aligned in mothur (version 1.36.1[50]) using the SILVA template[51] (SSURef_NR99_123), and the sequences were organized into clusters of representative sequences based on taxonomy called operational taxonomic units (OTU) using the UPARSE pipeline[52]. In the ferrets, all except two libraries generated more than 3000 reads per sample. A total of 649,440 sequences were subsequently clustered into 259 OTUs with a sequence similarity threshold of 97%[50], a length threshold of 250 bp and an expected error threshold of 0.15. For human samples, the distribution of reads per sample was much more variable, with an average of ~10,000 reads per sample. A handful of under-represented samples (below read threshold of 50 reads) were removed prior to the downstream analyses. A total of 2,300,072 sequences were sorted into 707 OTUs, using the same thresholds as above and the same down-stream filtering of the OTUs and samples was performed in a similar manner. Initial filtering of the samples ensured discarding samples containing less than five sequences. Libraries were normalized using metagenomeSeq's cumulative sum-scaling method[53] to account for library size acting as a confounding factor for the beta-diversity analysis. In addition to discarding singletons, OTUs that were observed fewer than five times in the count data were also filtered out to avoid the inflation of any contaminants that might skew the diversity estimates.

**Informatics**. Beta-diversity metrics were calculated across all samples using the Bray–Curtis dissimilarity index, and overall trends in the community composition for ferrets and humans on the basis of presence or absence of the flu infection were explored using Principal Coordinates Analysis (PCoA) in QIIME[54] (version 1.9.1) and then visualized in Emperor[55] (version 0.9.51).

Taxonomic classification of the samples was done by classifying the representative sequences from the OTUs using mothur and the SILVA database, with a confidence threshold of 97%. The relative abundances for the taxonomic profiles for each subject was calculated in QIIME using summarize_taxa.py. The visualization of the top ten most prevalent taxa for each of the organisms was done in R (version 3.2.2) using dplyr and reshape2 to manipulate the data and ggplot2 for generating the plots. Following the qualitative analysis of the data, we employed an infinite dimensional generalization of the multinomial Dirichlet mixture model (iDMM)[22], which tries to model the original set of communities with the input data with additional posterior-predictive probabilities (PPD) for statistical cutoffs. The model was executed over 1000 iterations for all ferrets and 2000 iterations for all human patients (regardless of infection state) since this parameter should increase with the number of samples present in the data set. Scripts located at https://github.com/jacobian1980/ecostates were improved by introducing a seed in the beginning of the algorithm to improve the reproducibility of the model and optimized the community number based on the PPDs, which compare empirically observed data with the data that would be expected if the DMM were the correct underlying model[56,57]. All downstream analyses with the communities, including exploration of community membership, were performed in R. In addition, a diagnostic OTU was computed for each ecostate, or sampled community, which is the OTU with the highest posterior-predictive probability in the ecostate and therefore drives the clustering. The quantitative portion of the analysis was supplemented by performing random-forest classification on the data to confirm the diagnostic results using Scikit-Learn (version 0.18.1) in Python (version 3.5.2) from Continuum Analytics Anaconda Suite. The training data set included: a *(n × m)*-dimensional attribute matrix consisting of the relative abundance values for the OTUs and the samples, where n and m refer to the number of samples and the number of OTUs, respectively, and a *(n)*-dimensional vector relating each observation to the two experimental states (positive and negative for the virus). The average of the feature importance vectors from 20,000 models that could accurately

predict all five left-out samples (~85% accuracy) was computed to obtain a weight for each OTU's predictive capacity to classify the experimental state of each sample. The hyperparameters for the random-forest model were 618 decision trees per forest, gini index as impurity criterion, and the square root of the number of features (OTUs in this case) to use for each split in the decision tree.

To further investigate the phylogenetic placement of the infected OTUs observed in our study among known Pseudomonas/Acinetobacter strains found in NCBI, we analyzed the infected OTUs (Otu000002_human, Otu000003_human and Otu000004_ferret) using BLAST against the "16S ribosomal RNA sequences (Bacteria and Archaea)" database and picked the top 25 reference hits that aligned with 100% coverage to each query sequence. Additionally, a few Euryarchaeota strains were also included as the outgroup. These reference sequences were then trimmed to extract the V1–V3 region, aligned to the query OTU sequences using clustalo (v 1.2.1), followed by phylogenetic tree building using RAxML (v 8.1.20) for 100 bootstrap iterations.

**Reporting summary**. Further information on research design is available in the Nature Research Reporting Summary linked to this article.

## Data availability
Raw amplicon sequence reads for this study have been deposited to Sequence Read Archive (SRA) under accession number: SRP009696 [BioProject accession number: PRJNA76689] for the ferrets and accession numbers: SRP092459 [BioProject accession number: PRJNA240559] and SRP128464 [BioProject accession number: PRJNA240562] for the infected and uninfected human subjects, respectively.

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

## Acknowledgements

The authors would like to thank research nurse Claudia Marco and the following clinical fellows and pediatricians that contributed to the recruitment of patients and the collection of samples used in this study: Marta Aravena, Catalina Gutierrez, Tania Lopez, Regina Perez, and Cecilia Vizcaya from the Department of Pediatric Infectious Diseases and Immunology, Facultad de Medicina, Pontificia Universidad Católica de Chile, Santiago, Chile. This project has been partly funded with federal funds from the National Institute of Allergy and Infectious Diseases, National Institutes of Health, Department of Health and Human Services under Contract Number HHNS272200900007C/ HHSN266200700010C and Grant Number U19AI110819 (to JCVI), grants from the Comisión Nacional de Investigación Científica y Tecnológica (FONDECYT 1121172 and 1161791 to R.A.M.; and PIA ACT 1408 to R.A.M. and M.F.), and the Chilean Ministry of Economy, Development and Tourism (P09/016-F to R.A.M.). This study was also partially supported by CRIP (Center for Research in Influenza Pathogenesis), an NIAID funded Center of Excellence for Influenza Research and Surveillance (CEIRS, contract # HHSN272201400008C) and by NIAID grant U19AI135972 (to A.G.-S. and R.A.M.).

## Author contributions

D.K. and R.R. analyzed the data, prepared illustrations, and wrote the paper. M.F. designed human cohort study, recruited patients, collected clinical metadata, and wrote parts of the paper. G.S.T. and B.E.P. carried out data analysis and wrote parts of the paper. A.B. carried out data analysis, prepared illustrations, and wrote parts of the paper. D.W. and B.M. obtained funding, designed and supervised experiments, and analyzed the data. S.D. supervised experiments and analyzed the data. I.B. recruited patients and collected clinical metadata. R.A.H. performed sequencing experiments and metadata compilation. M.S., I.M., and R.A.A. performed ferret experiments. I.S. performed data processing and analysis. K.E.N. obtained funding, supervised this study, and wrote parts of the paper. A.G.-S. conceived and supervised this study, and wrote the paper. C.L.D. supervised this study, designed informatics analyses, analyzed the data, prepared illustrations, and wrote the paper. R.A.M. obtained funding, conceived and supervised this study, designed and performed experiments, analyzed the data, prepared illustrations, and wrote the paper.

## Competing interests

The authors declare no competing interests.
