## [Peer Review File · Nature Communications]

Reviewers' comments:

Reviewer #1 (Remarks to the Author):

In this manuscript, Kaul et al investigate modulation of the upper respiratory tract microbiome following influenza A virus infection in both humans and ferrets. The authors identify perturbations in the microbiome detected between both species, supporting the need to consider bacterial co-infections in the context of viral infection and further illustrating that data generated in the ferret model can closely recapitulate what occurs during human infection. Research investigating the role the human URT microbiome plays in influenza virus infection, and levels of potential similarity between ferret and human microbiomes, represent important and understudied areas in the field. This well-written article represents an ambitious first step in addressing these questions, and would be of high interest to the field. However, there are numerous areas of the manuscript which require additional clarification, justification, and modification.

Major comments:

1. It is unclear why out of the n=30 human patients in this study, 2 were diagnosed yet not confirmed as influenza positive, yet were nonetheless included in the flu-positive group (page 20 lines 434-5). Considering the authors state in the introduction that “temporary disturbances to the microbiome due to the changes in the local epithelia during acute or chronic conditions has also been reported... (page 4 lines 75-6)”, how are the authors confident that these two patients truly meet inclusion criteria for this study (e.g. what was the definition for diagnosis)? Information pertaining to the prior influenza vaccination history of these patients is also missing. Furthermore, were these human patients tested for co-infection with other viral respiratory pathogens? The methods state that control patients were confirmed negative for “13 other common respiratory viruses (page 21 line 440)” but it is unclear if patients in the infected group were similarly assessed.

2. While inclusion of ferret data in this study is of benefit, there are areas where this data must be better contextualized, qualified, or presented. It is unfortunate that, per page 12 lines 258-60, ferret experimental infection and control experiments occurred at different times, potentially contributing to divergent microbiomes between both groups at the onset of the study (as shown in time 0 timepoint in figure S7). Furthermore, as the methods state that virus inoculated ferrets were administered 5ml of virus whereas control animals were administered 0.5ml (page 22 line 470), this either represents a typographical error or a poor “control” for this atypical inoculum volume. Did the authors investigate if administration of a liquid intranasally-administered inoculum alone perturbed the ferret URT microbiome? As Figure 3 shows differences in beta diversity among experimentally inoculated ferrets 1hr post-inoculation, well before a full viral life cycle could occur, it would be

helpful if the authors could address the potential contribution of the inoculation method employed towards perturbing the URT microbiome.

3. The authors show in Figure 4C and 4D “the community composition for one representative influenza-infected and one uninfected ferret” (page 13 lines 272-3). However, the “representative” influenza-infected ferret shown in this main text display image has a baseline URT microbiome profile that is representative of only 3/7 ferrets in this group (per data presented in figure S7). It is misleading for the authors to declare a ferret representative when <50% of the ferrets in this group share this profile at t=0. The authors must be more upfront about employing a batch of ferrets which possessed a divergent baseline profile of pseudomonadales and clostridiales, or expand Figure 4 to show two representative ferrets, one from each group where taxonomic profiles differ. The ultimate conclusions drawn by the authors from this data (lines 274-287) that pseudomonadales abundance increases during acute infection, is still supported by the data; the authors just need to be more upfront and revealing about the level of diversity in their challenge experiment so readers can interpret this data fully.

Minor comments:

1. Page 6, line 121: Please include the reference supporting the “Anna Karenina” principle of microbiomes here, not just in the discussion (currently ref 29 in discussion).
2. Page 6, line 130/page 7, line 132: the section states that that the human URT microbiome is not dependent on influenza virus subtype, but the section text does not provide any data to support this conclusion. Please justify this conclusion within the main body text. Considering other research has found that influenza viruses can differentially interact with bacterial pathogens during infection (as reviewed in McCullers Nat Rev Immunol PMID 24590244), the authors should be more precise in their wording here and more upfront about only using patients from one influenza season and ferrets infected with only one virus subtype to draw their overall conclusions.
3. Figure 2, Figure S3: please provide additional text/labels in the figure legend text describing what is presented on the x axis. It is unclear if each vertical line represents one unique sample, and if so, how the samples are grouped (by person/animal? By collection date? Etc). Not all readers are familiar with this data presentation so additional clarity is needed.
4. Figure 4B, please state in the figure legend or methods the limit of detection for the titration method shown, and modify this figure to reflect that limit of detection (it is unlikely the LOD is indeed 0 pfu as presented in this graph).

5. In the methods, please include information regarding the health status of ferrets pre-experiment (vaccine history, pathogens that are monitored pre/post animal use in the laboratory).

6. When discussing “potential reasons for the discrepancy between ferrets and humans (page 19 line 399)”, the authors should also note that the challenge route and dose for ferrets (large volume and high challenge titer) is not representative of human exposure and/or infection. Conducting this type of comparison to human data employing an aerosol challenge or via use of contact ferrets that were infected via a more “natural” route might offer a more representative experimental model for the comparison the authors are trying to draw in this study.

Reviewer #2 (Remarks to the Author):

The authors present the results of two studies. The first compares the nasopharyngeal microbiome of 30 patients hospitalized with influenza like illness (28 with confirmed influenza iehter H1N1pdm09 or H3N2) and 22 healthy individuals confirmed negative for influenza A and 13 other common respiratory viruses – selection criteria for controls not stated. The second study is of ferrets, 7 infected H1N1 pandemic strain through intranasal inoculation and 7 control animals infected with sterile PBS. Although the study is justified in the introduction by the statement ‘the effect of IAV replication and induction of innate immune response on the composition of the human or animal URT microbiome remains to be elucidated and analyzed in depth on a community wide scale.’ The study does not directly address innate immune response. The major take-home messages are that ferrets are a good model for modulation of the microbiome in humans, and that microbiome disturbance and resilience dynamics may be critical to addressing the bacterial co-infections associated with influenza-derived morbidity. There is no data directly supporting this second conclusion.

Although this study has many strengths, there are several areas of concern.

First, authors should review and include reference to more recent articles on this topic:

Ramos-Sevillano E et al. CID 2019:68

Wouter AA de Steehuijsen Piters et al Nature Communications 2019

It is surprising that microbiome data are not classified to genus species level. This choice should be justified.

The paper would be enhanced by further exploring the effects of age. As shown in the work by Lee et al. (and others) nasopharyngeal microbiota vary by age. It would be helpful to present results separately for children from adults. Depending on the distribution <2, 2 to <6, 6 to 18, 18 to 64 and 65 and old would be very helpful. The groupings in Table S1 could be made more granular.

There is a suggestion in the literature that the microbiome response may vary by infecting influenza type. It would be helpful if the authors commented on any changes by type in the human study. Similarly, as there are two cases without confirmed influenza. I would suggest excluding them or at a minimum, including a sensitivity analysis.

The data could be presented in a way that makes more clear the dynamics. For example, showing ecostates at each time point for individuals or probability of changing between ecostates between time points (perhaps using a Markov model) as others have done.

The finding of an association with *Pseudomonas* is surprising. This result should be compared to the literature. This apparent bloom is not consistent with previous studies. Have they used Dada2 or equivalent program to determine species? *Pseudomonas* is a large order. The conclusion that there was no difference in microbiome elasticity by age, sex or antibiotic usage is not supported by the data presented. It seems unlikely that they have sufficient power to test the effects of age

The finding that microbiomes change in response to the IAV infection is not new (see suggested references).

The comment that “increased abundance of Pseudomonadales and decreased abundance of Clostridiales and Actinobacteria suggest a potential use for probiotic treatments” should be removed. These data are compositional.

Minor:

Methods: please provide information on the inclusion of mock communities in with sequencing. Note that the Qiagen All Prep Kit will result in an under representation of Gram positive organisms (which might be commented on in the discussion).

The authors might consider comparing the taxa by infection state also using ALDEX2 for comparison to the Random Forest analyses.

Is p value for ANOSIM on line 139 p 7 correct? Appears to be wrong as the R value is very low.

Figure 2 is described in the text as giving taxa at the order level but is actually presenting at the class level. The description in the text is particularly confusing as the text talks about phyla and order level and equates Gammaproteobacteria to Pseudomonas.

Suggest substitute Figure 3S for Figure 2 or modify Figure 2 so reader NOT looking at the supplement can see that most of the Gammaproteobacteria were Pseudomonas. Currently this is a bit disingenuous.

P. 4 line 206 – it is not clear to this reader why it is surprising that the diagnostic OTUs for all four ecostates for the human samples are also among the first 10 most abundant OTUs of the data, as those the ones most likely to drive the DMM.

P. 13 line 272 – how were the samples chosen for presentation in Figures 4C and 4D? Would be stronger if all were presented, or if only two are chosen, present the variation found between the samples.

p. 13 line 287 – In how human samples were basal levels re-established? If mentioned here, this analysis should appear in the analysis of the human study.

Figure S2. I found this figure confusing. What were the posterior probabilities for assignment? Were the DMM run including all human samples, regardless of infection status? Or separately by infection status? Please clarify here and in the methods.

Figure S4. It seems that this only presents results for 4 influenza infected subjects and 2 healthy subjects? Why not all? How were these selected?

Reviewer #3 (Remarks to the Author):

In this interesting study, the authors have collected time-series nasopharyngeal swabs or nasal washes to analyze the upper respiratory tract microbiota in a human cohort and in ferrets during influenza A virus infection. The most important aspect of this study is that it aims to explore the dynamics of the microbiome over time after disturbance with a viral pathogen. Access to this type of data in human cohorts is of real value and not always easy. The main observation is that *Pseudomonas* appears to be the taxon most clearly associated with disease, and this is seen in both humans and ferrets. While this is not novel per se, as other studies have reported the association of *Pseudomonas* with flu infection, the fact that *Pseudomonas*/*Pseudomonadales* increase in relative abundance over the course of the infection and then decrease when the infection is cleared is an interesting observation.

There are, however, a few problems with the study as it currently stands:

1. (a) The control samples for the human cohort are a bit of an issue. The majority of the flu positive samples were from individuals that were hospitalized (based on Table S1); it is not clear where the control samples were collected - community clinic, hospital, home, etc.

(b) There should be more details as to whether the samples from the flu-positive patients were all collected while the patients were in the hospital or not.

The environment can have a drastic effect on the composition of the respiratory microbiome, and this is particularly true for *pseudomonas*.

(c) The flu samples were collected over the course of 2 years – there are no details as to which seasons -- while the controls were collected 2 years later and all mostly in the fall season (March to June 2014 in the southern hemisphere). *Pseudomonas* is known to be seasonal. This may be difficult to do at this point, but ideally a few control samples from individuals that are matched for month/year with the flu-positive (even if not longitudinal sampling) would help. Without that, the clustering of the data as seen in Fig. 1 – which is the basis for most of the follow-up analyses -- could just be due to confounding factors associated with different environmental microbes circulating at the time samples were collected. The ferret data redeem this limitation a bit because the same trend is observed, but the number of animals is a bit low.

2. (a) It not clear to me why the 2 hospitalized patients that did not test positive for flu would be included. The rationale stated in the methods is that “they displayed a perturbation of their microbiome so they were included”. This is a circular argument.

(b) NP swabs were collected from 30 hospitalized individuals but NPs from 33 infected subjects underwent microbiome analysis (line 452). Who are these 3 patients?

3. It seems like a missed opportunity that the human time series data were not more fully analyzed, like was done for the ferret data. At a minimum, an analysis of whether samples cluster by time-points would be interesting – something a bit like Fig. 3.

4. For both the human and ferret data there could also have been some statistical analyses of the dynamics over time to get a more robust measure of taxa trends over time. The results as presented focus on showing relative abundance graphs at different timepoints, and mostly only for representative samples. There are spline-based tools, which also correct for issues such as different time points for each subject, that would be interesting to try. QIIME 2 actually also has a plug-in for longitudinal analyses.

5. The data analysis needs a bit of clarification. How many of the human samples had a reasonable number of reads? For the ferret data, the authors state that only 2 of the libraries did not have more than 3000 reads. For the human data, the only statement is that the distribution of reads per sample was more uneven. There needs to be more explicit information to better gauge the quality of the data.

There should also be some description of how many sequence runs were done for each study, and whether there are potential batch effects.

6. Line 297 should say “Table 3” not “Table 2”.

Point-by-point response to referee's comments: manuscript NCOMMS-19-14037

Reviewers' comments:

Reviewer #1 (Remarks to the Author):

In this manuscript, Kaul et al investigate modulation of the upper respiratory tract microbiome following influenza A virus infection in both humans and ferrets. The authors identify perturbations in the microbiome detected between both species, supporting the need to consider bacterial co-infections in the context of viral infection and further illustrating that data generated in the ferret model can closely recapitulate what occurs during human infection. Research investigating the role the human URT microbiome plays in influenza virus infection, and levels of potential similarity between ferret and human microbiomes, represent important and understudied areas in the field. This well-written article represents an ambitious first step in addressing these questions, and would be of high interest to the field. However, there are numerous areas of the manuscript, which require additional clarification, justification, and modification.

Major comments:

1. It is unclear why out of the n=30 human patients in this study, 2 were diagnosed yet not confirmed as influenza positive, yet were nonetheless included in the flu-positive group (page 20 lines 434-5). Considering the authors state in the introduction that “temporary disturbances to the microbiome due to the changes in the local epithelia during acute or chronic conditions has also been reported... (page 4 lines 75-6)”, how are the authors confident that these two patients truly meet inclusion criteria for this study (e.g. what was the definition for diagnosis)? Information pertaining to the prior influenza vaccination history of these patients is also missing. Furthermore, were these human patients tested for co-infection with other viral respiratory pathogens? The methods state that control patients were confirmed negative for “13 other common respiratory viruses (page 21 line 440)” but it is unclear if patients in the infected group were similarly assessed.

Response: *The 2 individuals noted were diagnosed clinically by their attending physician, on the basis of presenting to the hospital with influenza-like-illness (ILI); we have clarified this in the Methods. Nevertheless, we were not able to confirm their diagnosis by a diagnostic test. We appreciate and agree with the reviewer's comment that including these 2 individuals in the study might create confusion and the possibility of unintentionally introducing background noise to our analyses. Hence, we have excluded the data from these 2 individuals in the analyses. All statistical tests, tables, and figures have been updated.*

We have also included now the data on vaccination for each group (Table S1). Only 6 individuals had been vaccinated against influenza in the last 12 months, and for 1 individual this data was not available. We performed beta diversity, ANOSIM, and ADONIS analyses to evaluate if the clustering and/or the association of the microbiomes could be explained by vaccination (new Fig. S2). We found that there was correlation in the vaccinated individuals,

however; there was little indication of a real grouping in the ordination plot. Finally, we have also clarified now that all the infected individuals were also tested against 13 other pathogens. In only 2 of the individuals we detected co-infections at the beginning (day of recruitment), but who were positive to influenza and negative to other viral pathogens in subsequent test. This has been noted in the revised text (lines 450-452).

2. While inclusion of ferret data in this study is of benefit, there are areas where this data must be better contextualized, qualified, or presented. It is unfortunate that, per page 12 lines 258-60, ferret experimental infection and control experiments occurred at different times, potentially contributing to divergent microbiomes between both groups at the onset of the study (as shown in time 0 timepoint in figure S7). Furthermore, as the methods state that virus inoculated ferrets were administered 5ml of virus whereas control animals were administered 0.5ml (page 22 line 470), this either represents a typographical error or a poor “control” for this atypical inoculum volume. Did the authors investigate if administration of a liquid intranasally-administered inoculum alone perturbed the ferret URT microbiome? As Figure 3 shows differences in beta diversity among experimentally inoculated ferrets 1hr post-inoculation, well before a full viral life cycle could occur, it would be helpful if the authors could address the potential contribution of the inoculation method employed towards perturbing the URT microbiome.

Response: *The sentence in page 12, lines 258-60, was poorly written, as we intended to say that in previous analyses of uninfected ferrets, we had found an intrinsic variation in the microbiome composition on each animal due to a high degree of personalization and because these are outbred animals. We have amended the text to reflect this better (lines 260-262). Therefore, while we acknowledge that some variation in the microbiome could be contributed by the experimental setting, we clarify that the experiments with infected and uninfected ferrets were done simultaneously. Hence, no specific variations at the beginning of the experiment (day 0) could be attributed due to experimentation, animal handling or sample processing, but instead any initial variation is most likely due to individual animal differences and to the infection dynamics on each animal.*

The discrepancy in the volume of virus administered was an involuntary typographical error that we have amended (it was 0.5 mL for both case and control animals, line 488). Moreover, indeed, in a pilot experiment conducted with uninfected we compared intranasal inoculation and the effect of nasal washes and nasal swabs taken from the same individual using 2 animals at 2 time points (two days apart). We obtained comparable results for either the swab or wash samples, yielding comparable results, with washes showing more consistency in assessing diversity. We also observed great consistency in regards to abundance and diversity in samples from the same inoculated animal at the two time points (e.g. no specific dynamic changes after a nasal wash or a swab), but greater variation (or diversity) was seen between the animals. This is the data that supported the idea of an intrinsic difference in the basal microbiome of each outbred animal stated above. Similarly, the vehicle (PBS), inoculation volume and route is the same for both the uninfected and the infected animals. Hence, this is the best control for this experiment, since any underlying change in the microbiome resulting from this procedure would be the same as this was done concurrently in all animals. Again this supports the notion that any changes are due to basal microbiome levels of each animal and due to infection dynamics.

3. The authors show in Figure 4C and 4D “the community composition for one representative influenza-infected and one uninfected ferret” (page 13 lines 272-3). However, the “representative” influenza-infected ferret shown in this main text display image has a baseline URT microbiome profile that is representative of only 3/7 ferrets in this group (per data presented in figure S7). It is misleading for the authors to declare a ferret representative when <50% of the ferrets in this group share this profile at t=0. The authors must be more upfront about employing a batch of ferrets which possessed a divergent baseline profile of pseudomonadales and clostridiales, or expand Figure 4 to show two representative ferrets, one from each group where taxonomic profiles differ. The ultimate conclusions drawn by the authors from this data (lines 274-287) that pseudomonadales abundance increases during acute infection, is still supported by the data; the authors just need to be more upfront and revealing about the level of diversity in their challenge experiment so readers can interpret this data fully.

Response: *We agree that the baseline microbiome (T0) in ferrets displays a level of diversity, as discussed above. Nevertheless, this has been addressed in the same figure by the variability in Pseudomonadales levels shown at the earlier time points (Fig 4E). This is primarily why we showcase the taxa trend for only one of the ferrets, chosen at random, in the main text figure, and the same for all ferrets as supplemental data. To be more precise, we have deleted the word “representative” from the main text. The main conclusion from the figure is the increase in Pseudomonadales over the acute timepoints and this is still supported by the data as presented, which has also been pointed out by the reviewer, hence including two representative ferrets would be redundant since this is already shown in Supplementary Fig. 7 (now Fig. S9).*

Minor comments:

1. Page 6, line 121: Please include the reference supporting the “Anna Karenina” principle of microbiomes here, not just in the discussion (currently ref 29 in discussion).

Response: *Reference has been added.*

2. Page 6, line 130/page 7, line 132: the section states that that the human URT microbiome is not dependent on influenza virus subtype, but the section text does not provide any data to support this conclusion. Please justify this conclusion within the main body text. Considering other research has found that influenza viruses can differentially interact with bacterial pathogens during infection (as reviewed in McCullers Nat Rev Immunol PMID 24590244), the authors should be more precise in their wording here and more upfront about only using patients from one influenza season and ferrets infected with only one virus subtype to draw their overall conclusions.

Response: *We appreciate the reviewer’s comment. Hence, we performed an ADONIS test to evaluate the effect of subtypes on the microbiome, which was found to be significant but showed little indication of a real grouping in the ordination plot (Fig S2). Hence, the effect cannot be determined with this data set. We have amended this statement (lines 156-157).*

3. Figure 2, Figure S3: please provide additional text/labels in the figure legend text describing what is presented on the x axis. It is unclear if each vertical line represents one unique sample,

and if so, how the samples are grouped (by person/animal? By collection date? Etc). Not all readers are familiar with this data presentation so additional clarity is needed.

Response: *As noted in the legend, this data only represents human subjects, with the healthy cohort on the left and the infected cohort on the right. The host has been identified in the legend text. Each vertical column represents a unique sample (clarified in the revised legends), and the sample labels are deliberately omitted for presentation (they would be illegible). Taxonomic groups have been sorted by most prevalent taxa at class level (Gammaproteobacteria in the infected cohort and Actinobacteria in the healthy cohort) present among the samples, on account of total relative abundance across all samples.*

4. Figure 4B, please state in the figure legend or methods the limit of detection for the titration method shown, and modify this figure to reflect that limit of detection (it is unlikely the LOD is indeed 0 pfu as presented in this graph).

Response: *We have changed the graph to reflect the assay limit of detection, which was 10 pfu/ml. This was also included in the figure legend.*

5. In the methods, please include information regarding the health status of ferrets pre-experiment (vaccine history, pathogens that are monitored pre/post animal use in the laboratory).

Response: *The ferrets were purchased from Triple F farms, a company that provides influenza-free and specific pathogen free animals that have not been vaccinated against influenza. Hence the animals are certified to be free of common bacterial and viral infections including, a number of bacteria (*Salmonella*, *Staphylococcus*, *Mycoplasma*, *Streptococcus*, among others) and various viruses including multiple influenza subtypes, rotavirus and coronavirus, among others (for a full list see <http://www.triplefresearch.com/influenza-free-ferrets/> and <http://www.triplefresearch.com/downloads/>). As stated in the methods, prior to purchase, we also re-tested the animals to confirm they have not been infected by seasonal influenza. Once in the animal facility, the animals were housed individually in poultry isolators that are fitted with “High Efficiency Particulate Air (HEPA) filters that provide filtered air to the cages. HEPA filters remove—from the air that passes through—at least 99.97% (by US standard) of particles with a diameter $\geq 0.3 \mu\text{m}$. In addition, at all times the ferrets are only handled by personnel wearing an N95 mask, which also prevents the transmission of airborne pathogens. For sample collection the animals are handled inside a laminar flow biosafety cabinet that provides clean HEPA filtered air and prevents the animals to become in contact with the out side air. Hence, the overall procedure and infrastructure provides the adequate conditions to keep the ferrets “pathogen free” throughout the experiment, except for the influenza virus inoculation administered to them intranasally. We have included these data in the methods section (lines 479-485).*

6. When discussing “potential reasons for the discrepancy between ferrets and humans (page 19 line 399)”, the authors should also note that the challenge route and dose for ferrets (large volume and high challenge titer) is not representative of human exposure and/or infection. Conducting this type of comparison to human data employing an aerosol challenge or via use of contact ferrets that were infected via a more “natural” route might offer a more representative experimental model for the comparison the authors are trying to draw in this study.

Response: We have added a statement to this effect (lines 410-413).

Reviewer #2 (Remarks to the Author):

The authors present the results of two studies. The first compares the nasopharyngeal microbiome of 30 patients hospitalized with influenza like illness (28 with confirmed influenza either H1N1pdm09 or H3N2) and 22 healthy individuals confirmed negative for influenza A and 13 other common respiratory viruses – selection criteria for controls not stated. The second study is of ferrets, 7 infected H1N1 pandemic strain through intranasal inoculation and 7 control animals infected with sterile PBS. Although the study is justified in the introduction by the statement ‘the effect of IAV replication and induction of innate immune response on the composition of the human or animal URT microbiome remains to be elucidated and analyzed in depth on a community wide scale.’ The study does not directly address innate immune response. The major take-home messages are that ferrets are a good model for modulation of the microbiome in humans, and that microbiome disturbance and resilience dynamics may be critical to addressing the bacterial co-infections associated with influenza-derived morbidity. There is no data directly supporting this second conclusion.

Although this study has many strengths, there are several areas of concern.

First, authors should review and include reference to more recent articles on this topic:

Ramos-Sevillano E et al. CID 2019:68

Wouter AA de Steehuijsen Piters et al Nature Communications 2019

Lee KH et al. PLoS One 2019 Jan 9;14(1):e0207898

Response: We acknowledge and thank the reviewer for pointing out these publications, 2 of which came out while our paper was under revision. We have added the Ramos-Sevillano reference to the discussion, though it must be noted that they did not observe microbiome changes. They also sampled the oropharyngeal microbiome, which is distinct from the nasopharyngeal microbiome. Wouter AA de Steehuijsen Piters uses an attenuated vaccine in healthy non-smoking adults and thus, it is not comparable given that this attenuated virus does not replicate efficiently in the respiratory tract. Our cohort represents a natural infection with seasonal viruses that resulted in a severe infection. Lee et al, conducted a contact follow up study that used 2 samples per patient that determined that microbiome at the time of enrollment and at the last day of follow up (median time between samples was 9 days). Hence, this was not a temporal study like our own and thus is not relevant here, except for the fact that they saw some changes in what they call “Community state types (CST)” between time points, which agrees with our findings. We have included this reference to acknowledge this fact. All these has been included on lines 360-367. Of note, we have also included references to 2 recently published manuscripts describing microbiome studies in humans (Lines 350-352). One was a follow up study of the household contact study described above (Tsang, T. K. et al. Clin Infect Dis, 2019 Sept doi:10.1093/cid/ciz968), and the second manuscript (ding Ding, T. et al. MBio 10, 2019 Jul doi:10.1128/mBio.01296-19) reports the analyses of the microbiome composition according to vaccination and Influenza subtype. Both studies use nasopharyngeal samples obtained at single timepoints during acute influenza infection, and are then compared with the microbiome of healthy individuals. Again, these studies differ drastically from ours and hence

they are not comparable.

It is surprising that microbiome data are not classified to genus species level. This choice should be justified.

Response: *Taxonomy for the OTUs derived from the microbiome data for both humans and ferrets has been resolved down to the genus level for almost all, and species level when possible, given the length threshold (250 bp) for the OTUs identified in our data. The reason we chose to show the taxa breakdown at the class and order level in Fig. 2 and Fig. 4, respectively, is for the readers to easily interpret the different taxa levels represented in the data. Including the genus/species level classification for all the OTUs would have overpopulated the plot, making it impossible to visualize. The entire OTU table is now included in the data source file allowing a reader to examine the data at finer taxonomic detail if desired.*

The paper would be enhanced by further exploring the effects of age. As shown in the work by Lee et al. (and others) nasopharyngeal microbiota vary by age. It would be helpful to present results separately for children from adults. Depending on the distribution <2, 2 to <6, 6 to 18, 18 to 64 and 65 and old would be very helpful. The groupings in Table S1 could be made more granular.

Response: *We agree that age is an important variable in microbiome evolution. We did test for the significance of effect of age on the microbiome dynamics in humans and concluded that while the p-value was significant, the clustering based on age was only moderately strong. Clustering the data points according to the distribution suggested by the reviewer would not have enough data points each grouping (i.e. for age < 2years) to make any sound statistical conclusions. Having said that, we have now added a supplemental figure to show the stratification of the microbiome on the basis of age groupings as suggested by the Reviewer (Fig S2).*

There is a suggestion in the literature that the microbiome response may vary by infecting influenza type. It would be helpful if the authors commented on any changes by type in the human study. Similarly, as there are two cases without confirmed influenza. I would suggest excluding them or at a minimum, including a sensitivity analysis.

Response: *We have performed the subtype analyses proposed and as stated above this has now been included in the text (lines 155-157). Also, as stated above, we have now excluded the two samples without confirmed influenza.*

The data could presented in a way that makes more clear the dynamics. For example, showing ecostates at each time point for individuals or probability of changing between ecostates between time points (perhaps using a Markov model) as others have done.

Response: *An example of the proposed visualization in the literature would be helpful (we have not found any examples of plots the reviewer describes). The iDMM algorithm employs MCMC to compute the posterior predictive probabilities.*

The finding of an association with Pseudomonas is surprising. This result should be compared to the literature. This apparent bloom is not consistent with previous studies. Have they used

Dada2 or equivalent program to determine species? Pseudomonas is a large order. The conclusion that there was no difference in microbiome elasticity by age, sex or antibiotic usage is not supported by the data presented. It seems unlikely that they have sufficient power to test the effects of age.

Response: *We have added phylogenetic inferences (DADA 2 is a clustering algorithm, not a phylogenetic identifier) to show that the humans were infected by a Pseudomonas of unidentifiable species (Supplemental figure 5). Analysis of the ferret data shows that they are infected by an Acinetobacter (a Pseudomonadales, Supplemental figure 5).*

The finding that microbiomes change in response to the IAV infection is not new (see suggested references).

Response: *To an extent, we disagree. We have added the Ramos-Sevillano reference to the discussion, though it must be noted that they did not observe microbiome changes and looked at a different microbiome (oropharyngeal). Wouter AA de Steehuijsen Piters uses an attenuated vaccine, which as previously mentioned it is not comparable. Lee et al showed changes in what they call “Community state types (CST)” between time 2 points, however; they did not show associative changes due to viral infection overtime. Instead, this is rather an observational report of the differences seen. In our study we used multiple samples obtained sequentially from the same infected individuals, and report the dynamic temporal changes of the microbiome. Additionally, while, we understand that Pseudomonas sp. has been associated with IAV infection in the past, these were all cross-sectional studies reporting a single time point in their analyses. Therefore, the finding that Pseudomonas sp. blooms at peak timepoints over the course of the infection and that it subsequently clears, has never been observed before and is what makes our study novel.*

The comment that “increased abundance of Pseudomonadales and decreased abundance of Clostridiales and Actinobacteria suggest a potential use for probiotic treatments” should be removed. These data are compositional.

Response: *This comment has been removed.*

Minor:

Methods: please provide information on the inclusion of mock communities in with sequencing. Note that the Qiagen All Prep Kit will result in an under representation of Gram positive organisms (which might be commented on in the discussion).

Response: *No mock communities were included for sequencing. Instead, one positive and negative amplification control were included in each pool (total of two controls). The positive control amplified and the negative did not amplify, as expected. We have also added the following the discussion “It must be noted that, while the DNA extraction method used could results in a slight underrepresentation of gram-positive bacteria, this would be consistent across both infected and healthy control samples (lines 374-376)”*

The authors might consider comparing the taxa by infection state also using ALDEX2 for comparison to the Random Forest analyses.

Response: *While we agree that ALDEX2 is a valuable tool, we already included the iDMM diagnostic OTU determination, which is augmented by the random forest analysis for increased*

confidence. Given that ALDEX2 uses DMMs, it would be redundant to our iDMM approach.

Is p value for ANOSIM on line 139 p 7 correct? Appears to be wrong as the R value is very low.

Response: Both values are correct. While the interpretation of the p-value is independent of the R-value, understanding how both of them contribute to the results is important. There are numerous cases when the R-value is low, and its corresponding p-value is also low. The R-value explains effect size (separation between groups) as opposed to the p-value, which explains statistical significance and is prone to becoming more significant as the number of samples increases. We understand that the interpretation for ANOSIM alone can be confusing. Thus, we have complemented the statistical analyses with ADONIS test, which is known to be more robust.

Figure 2 is described in the text as giving taxa at the order level but is actually presenting at the class level. The description in the text is particularly confusing as the text talks about phyla and order level and equates Gammaproteobacteria to Pseudomonas.

Response: The reviewer is correct: All the classifications present in Fig. 2 are at the class level. We corrected the main text to match the actual classification of the figure. Actinobacteria, Clostridia, Gammaproteobacteria, etc. all have phylogeny ranks of class (also in Fig. 2 legend, lines 762-764).

Suggest substitute Figure 3S for Figure 2 or modify Figure 2 so reader NOT looking at the supplement can see that most of the Gammaproteobacteria were Pseudomonas. Currently this is a bit disingenuous.

Response: We appreciate the comment. We have modified Figure 2 in the main text to include the Gammaproteobacteria breakdown, previously presented in Figure S3.

P. 4 line 206 – it is not clear to this reader why it is surprising that the diagnostic OTUs for all four ecostates for the human samples are also among the first 10 most abundant OTUs of the data, as those the ones most likely to drive the DMM.

Response: We intended to merely present the observation in the manuscript; we're not surprised by it, but we have also removed the prefix "Remarkably," from that sentence.

P. 13 line 272 – how were the samples chosen for presentation in Figures 4C and 4D? Would be stronger if all were presented, or if only two are chosen, present the variation found between the samples.

Response: The samples were chosen at random to make the visualization concise and 4 panels are easier to look at than 14. In the revised text a comprehensive taxa breakdown for all timepoints for all ferrets is also presented as Supplemental Fig. 10.

p. 13 line 287 – In how human samples were basal levels re-established? If mentioned here, this analysis should appear in the analysis of the human study.

Response: Basal levels could not be determined for humans and we removed that sentence (which is confusing even to us on a reread).

Figure S2. I found this figure confusing. What were the posterior probabilities for assignment?

Were the DMM run including all human samples, regardless of infection status? Or separately by infection status? Please clarify here and in the methods.

Response: *The Infinite Dirichlet multinomial Mixture Model (iDMM) was run on all human samples, regardless of infection state, which resulted in the distribution of the data points into four ecostates (3 ecostates representative of the infected state and 1 for the healthy state, Table 2). This has already been referenced at line 177-186 in the main text and has now been clarified again in the methods now (line 540).*

In short, this algorithm predicted the posterior probabilities at each iteration, for each OTU, for each sample; by comparing observed data with the data that would be expected if the model were the correct. These posterior predictive probabilities can be thought of as clustering weights, and were analyzed for the final iteration (1000th for ferret data and 2000th for human data) for all OTUs, clustered by the final ecostate assignments.

Figure S4. It seems that this only presents results for 4 influenza infected subjects and 2 healthy subjects? Why not all? How were these selected?

Response: *Patients were selected at random, but preference was given to ones with comprehensive temporal data points. We have presented results for 4 influenza infected subjects because we wanted to essentially focus on taxonomically analyzing two patients for each infection state: infected [subtype specific H1N1 and H3N2] and uninfected.*

Reviewer #3 (Remarks to the Author):

In this interesting study, the authors have collected time-series nasopharyngeal swabs or nasal washes to analyze the upper respiratory tract microbiota in a human cohort and in ferrets during influenza A virus infection. The most important aspect of this study is that it aims to explore the dynamics of the microbiome over time after disturbance with a viral pathogen. Access to this type of data in human cohorts is of real value and not always easy. The main observation is that *Pseudomonas* appears to be the taxon most clearly associated with disease, and this is seen in both humans and ferrets. While this is not novel per se, as other studies have reported the association of *Pseudomonas* with flu infection, the fact that *Pseudomonas*/*Pseudomonadales* increase in relative abundance over the course of the infection and then decrease when the infection is cleared is an interesting observation.

There are, however, a few problems with the study as it currently stands:

1. (a) The control samples for the human cohort are a bit of an issue. The majority of the flu positive samples were from individuals that were hospitalized (based on Table S1); it is not clear where the control samples were collected - community clinic, hospital, home, etc.

Response: *The control samples were obtained from healthy volunteers thorough the hospital's outpatient clinic. This has been included now in the methods section (line 456-457). And have included a footnote on Table S1 to clarify this.*

(b) There should be more details as to whether the samples from the flu-positive patients were all collected while the patients were in the hospital or not.

The environment can have a drastic effect on the composition of the respiratory microbiome, and this is particularly true for *pseudomonas*.

Response: *The samples of the influenza-positive individuals were collected at the hospital, and in some cases upon released from the hospital some samples were taken at the outpatient clinic or in the patient's household. While we thank the reviewer for pointing out the fact that environmental factor can contribute to the microbiome composition, our data and analyses indicate that the dynamic change in the microbiome upon influenza virus infection is mainly dictated by disease state and possibly due to host responses (immune responses) rather than environmental factors. As pointed out by the reviewer, pseudomonas can be associated to specific environments. We believe that we have addressed and acknowledge this in our discussion, as we have pointed out that pseudomonas have been specifically linked to nosocomial infections as a result respiratory support treatments in hospital settings (line 390-392).*

(c) The flu samples were collected over the course of 2 years – there are no details as to which seasons -- while the controls were collected 2 years later and all mostly in the fall season (March to June 2014 in the southern hemisphere). Pseudomonas is known to be seasonal. This may be difficult to do at this point, but ideally a few control samples from individuals that are matched for month/year with the flu-positive (even if not longitudinal sampling) would help. Without that, the clustering of the data as seen in Fig. 1 – which is the basis for most of the follow-up analyses -- could just be due to confounding factors associated with different environmental microbes circulating at the time samples were collected. The ferret data redeem this limitation a bit because the same trend is observed, but the number of animals is a bit low.

Response: *We agree with the reviewer that the exact period of collection for the infected individuals was incompletely detailed in the methods section. Therefore, we have now added that the samples from influenza-infected individuals were collected during the winter season in the Southern Hemisphere in years, 2011 and 2012 (lines 443-444).*

In regards to the use of controls from a different season, our data argues against a potential effect of seasonality or confounding factors associated with different environmental microbes circulating at the time of sample collection. Our data shows, that if we compare samples from the beginning of infection versus the end of infection in both human and ferrets, the samples look more similar to each other. In ferrets, samples before infection look a lot like samples after resolution, with a similar dynamics and characteristics that we see in humans. Similarly, our control samples taken in a different year look more similar to the samples from infected individuals after disease resolution (e.g. healthy ecostate). Altogether, this strongly suggests that it is unlikely that our results and analyses would have changed drastically by taking controls from the same year, or in a different setting. While we cannot exclude that a specific effect of the microbiome dynamics could be due to a specific season at the time of the study, the change in microbiome reported here, we see it in humans and also in ferrets infected with IAV at different times and different locations. Therefore, this argues strongly that it is not necessary to analyze new controls, because our data indicates that this is not a phenomenon that occurs in a single season in the Southern hemisphere, nor that it is an effect of the environment of where the samples were taken. We also point out, that the ferret experiments were done in an animal facility in the northern hemisphere. Hence, this would also argue against cofounding environmental factor being responsible for the microbiome dynamics and the classification of the “ecostates” reported. Additionally, taking single control samples from individuals matched for month/year might not give a complete picture of that season's bacteria circulation. Likewise, it

would be an extremely complex and expensive clinical set up to obtain an ideal control cohort in a single season. We thus believe there is not an ideal control for the human infection study, and that is why we performed the complementary ferret study.

In regards to the ferret experiment, the reviewer mentions that the number of animals is a bit low. However, this seems like an understatement for two main reasons. Firstly, the ferret data is strongly supported by all the statistical analyses performed, which strongly indicate the perturbation of the microbiome in a dynamic and temporal manner, and the “ecostates” indicate that pseudomonas is a predictor of the unhealthy state. Including more animals would unlikely show different results or conclusions. Secondly, and possibly more importantly, given these data, it is unlikely that any IACUC committee would approve the use of more animals for experiments that already show statistically valid results. It is important to point out that a typical experimental setting with influenza in ferrets includes only 3 animals per group. Hence, the use of 7 animals per group is already a high number for this animal model.

2. (a) It not clear to me why the 2 hospitalized patients that did not test positive for flu would be included. The rationale stated in the methods is that “they displayed a perturbation of their microbiome so they were included”. This is a circular argument.

Response: *These samples were removed from the analyses as previously stated and suggested by reviewer 1 and 2 to avoid introducing background noise to our analyses.*

(b) NP swabs were collected from 30 hospitalized individuals but NPs from 33 infected subjects underwent microbiome analysis (line 452). Who are these 3 patients?

Response: *This was a typographical error in the text, which has been corrected now. Only 28 infected subjects were analyzed for the study.*

3. It seems like a missed opportunity that the human time series data were not more fully analyzed, like was done for the ferret data. At a minimum, an analysis of whether samples cluster by time-points would be interesting – something a bit like Fig. 3.

Response: *There were few patients with comprehensive timepoints, most had missing or unaligned time points since the human study was not as controlled as the ferrets, which is a caveat of the human data addressed in the paper.*

4. For both the human and ferret data there could also have been some statistical analyses of the dynamics over time to get a more robust measure of taxa trends over time. The results as presented focus on showing relative abundance graphs at different timepoints, and mostly only for representative samples. There are spline-based tools, which also correct for issues such as different time points for each subject that would be interesting to try. QIIME 2 actually also has a plug-in for longitudinal analyses.

Response: *While we understand the view of the reviewer, we did not find that these tools provided novel insights or display images that represented the data in any better way than already shown.*

5. The data analysis needs a bit of clarification. How many of the human samples had a

reasonable number of reads? For the ferret data, the authors state that only 2 of the libraries did not have more than 3000 reads. For the human data, the only statement is that the distribution of reads per sample was more uneven. There needs to be more explicit information to better gauge the quality of the data.

There should also be some description of how many sequence runs were done for each study, and whether there are potential batch effects.

Response: *We have clarified this in the text (lines 514-517) where we have added the following:*

“For human samples, the distribution of reads per sample was much more variable, with an average of approximately 10,000 reads per sample. A handful number of under-represented samples (below read threshold of 50 reads) were removed prior to the downstream analyses.”

6. Line 297 should say “Table 3” not “Table 2”.

Response: *We thank the reviewer for pointing out this mistake. It has now been resolved.*

Reviewers' comments:

Reviewer #1 (Remarks to the Author):

In this revised manuscript, Kaul et al have improved the clarity and analysis of parts of the study, notably by excluding human samples that were not confirmed to be influenza virus-positive from their analyses. However, there are still areas of underlying concern throughout the manuscript.

Comments:

Line 106, 473, and Table S1: the authors indicate nasal swab sample collection times as “days post-infection” but do not define what t=0 is for human cases. Time of entrance to hospital? Time of positive influenza virus diagnosis? Authors must either define what “post-infection” means here, or alter the timing descriptor to more accurately state what t=0 is, since unless these human subjects were infected experimentally, they cannot for sure know what the day of infection was.

Figure S7, the analyses of microbiome diversity among virus-infected ferrets is based on the assumption given by the authors, “since there are only two possible infection states for the ferrets, i.e. uninfected and infected...” (lines 903-904). However, researchers who employ influenza virus-infected ferrets frequently distinguish between the acute phase of virus infection (i.e. while the virus is being shed to detectable titer by infected ferrets/innate immune activation responses) and the convalescent phase of virus infection (i.e. while the animals have cleared virus/adaptive immune responses are taking place). Data from the authors support this as the beta diversity analysis shown in Figure 3 clearly shows differences between ferret samples collected between 1-7 p.i. and samples collected day 14 p.i. Why did the authors not further parse this data to distinguish between these two infection states as they did in Figure 3?

Lines 263-4. It is unclear why the finding that 4/7 ferrets exhibited “healthy” microbiomes prior to infection is touted by the authors as “remarkable.” Rather, it is perplexing to the reader why 3/7 ferrets from the infected group had divergent profiles (as supported in Figure 3 (lilac dots) and Figure S9). Furthermore, the wording is still unclear in this section if the “separate “healthy” experiment” described on line 260 is the same baseline experiment as what is referred to on line 264 (“...from an independent experiment”).

Figure 3, specify the legend label “infected others” in the text to more clearly state that these are samples collected days 1-7 p.i. from infected ferrets.

Lines 481-2, can the authors specify how long the animals were held at the institution prior to use in their experiments in the HEPA-filtered incubators? As the authors do not appear to have independently confirmed that all ferrets were seronegative on day 0, but rather relied on serological data based on pre-shipment potential exposures, it would be beneficial for the reader to know if sufficient time would have elapsed post-shipment where potential exposure to influenza virus (or a different pathogen) may have occurred to explain the divergent profiles observed among 3/7 infected-group ferrets as evidenced in their day 0 microbiome profiles.

Figure 4, it is still very unclear why the authors are showing microbiome data from an infected ferret that reflects a pre-infection baseline matching <50% of the ferrets in the infection group and <25% of all ferrets in the study. While the authors removed the word “representative” from the main text (yet, neglected to modify the accompanying y axis for panels C and D), it still feels disingenuous to the reader (who is less likely to access the full supplemental data files) to present data in a main display image that is not truly representative of the findings of the study. Ferret 595 shown in Figure 4 appears to have the highest collective peak abundance of pseudomonadales days 5 and 7 p.i. compared with all 7 ferrets in the virus-infected group. Selected data shown in display images should be truly representative of the dataset as a whole, and it does not appear to this reader that the profile for 595 is truly indicative of the group either at day 0 or days 5-7 p.i.

Reviewer #2 (Remarks to the Author):

Overall the authors have done a thorough job of addressing my comments. It would be helpful to add, either in the text or in the supplement, a description of how many when the samples were collected from infected human cases. It seems disingenuous to just say multiple samples without specifying the average number per case and showing the distribution for comparison with those taken at specified times from controls. Similarly, it would be useful to show the age distributions by age of cases and controls. Looking at the PCAs I did wonder if the controls were a much narrower age range than cases (the table uses a VERY large category (ages 2 to 65) which includes the majority of cases and controls.

Reviewer #3 (Remarks to the Author):

The authors have adequately addressed all my comments.

Point-by-point response to referee's comments: manuscript NCOMMS-19-14037A

Reviewers' comments:

Reviewer #1 (Remarks to the Author):

In this revised manuscript, Kaul et al have improved the clarity and analysis of parts of the study, notably by excluding human samples that were not confirmed to be influenza virus-positive from their analyses. However, there are still areas of underlying concern throughout the manuscript.

Comments:

Line 106, 473, and Table S1: the authors indicate nasal swab sample collection times as “days post-infection” but do not define what t=0 is for human cases. Time of entrance to hospital? Time of positive influenza virus diagnosis? Authors must either define what “post-infection” means here, or alter the timing descriptor to more accurately state what t=0 is, since unless these human subjects were infected experimentally, they cannot for sure know what the day of infection was.

Response: We appreciate the reviewer's comment, since he rightfully has pointed out that we had omitted to clearly define the timing of the samples from the human infected cohort. Therefore, we have revised the manuscript to indicate that the timing of the human samples was established from the time of onset of symptoms. Hence, we have made the following clarifications in the main text:

Line 100-101: “...and obtained nasopharyngeal swabs at multiple time points after the initial influenza-prompted hospital visits (days 1 to 37 after initial onset of symptoms)...”

Line 413: “...where zero time (Day 0) was the first day of symptom.”

Line 450-451: “Upon collection of all samples, the timing of the infected cohort samples was established as the time in days since the onset of symptoms..”

The Foot line for Table 1 was also and now indicates:

“^a All human and ferret samples were extracted from nasal washes and nasopharyngeal swabs, respectively, at several time points post symptom onset (humans) or post infection (ferret).”

Table S1 was also slightly modified to clarify the timing of the samples sequenced, thus it now indicates:

“...^a Days since onset of symptoms.”

Figure S7, the analyses of microbiome diversity among virus-infected ferrets is based on the assumption given by the authors, “since there are only two possible infection states for the ferrets, i.e. uninfected and infected...” (lines 903-904). However, researchers who employ influenza virus-infected ferrets frequently distinguish between the acute phase of virus infection (i.e. while the virus is being shed to detectable titer by infected ferrets/innate immune activation responses) and the convalescent phase of virus infection (i.e. while the animals have cleared virus/adaptive immune responses are taking place).

Data from the authors support this as the beta diversity analysis shown in Figure 3 clearly shows differences between ferret samples collected between 1-7 p.i. and samples collected day 14 p.i. Why did the authors not further parse this data to distinguish between these two infection states as they did in Figure 3?

Response: We agree with the reviewer that it would be valuable to analyze the data considering the two infection states as used in Figure 3. Hence, we performed the Diversity Distance Analyses by grouping the samples as infected T=0, Infected others (T=1,3,5,7 dpi), Infected T=14 and Uninfected Controls. The analyses supported the differences in diversity according to infection state as presented in Figure 3. Diversity within infected ferrets from T=14 (red) is the least, and diversity between all uninfected (U) and infected ferrets from the acute viral timepoints (T=1,3,5,7 dpi) is the highest, followed closely by diversity between all infection states (all Infected vs all Uninfected) being the second highest. Hence, we have now modified Figure S7 and made changes in the manuscript to reflect this update.

Line 254-256 now read:

“The t-statistic for the “All within infection” versus “All between infection” was -28.681 corresponding to a Bonferroni-corrected parametric p-value of 8.85e-161 (Table S5).”

Lines 263-4. It is unclear why the finding that 4/7 ferrets exhibited “healthy” microbiomes prior to infection is touted by the authors as “remarkable.” Rather, it is perplexing to the reader why 3/7 ferrets from the infected group had divergent profiles (as supported in Figure 3 (lilac dots) and Figure S9). Furthermore, the wording is still unclear in this section if the “separate “healthy” experiment” described on line 260 is the same baseline experiment as what is referred to on line 264 (“...from an independent experiment”).

Response: We agree with the reviewer that the wording of this sentence remained confusing. The part “...from an independent experiment” remained unchanged from the previous set of edits, and has now been deleted.

To clarify, the finding we thought was remarkable was the latter part of line 263, which is the fact that all 7/7 infected ferrets from T14 exhibited “healthy” microbiomes 14 days post infection. We had previously addressed the divergent profiles for the baseline microbiome (T0) among ferrets prior to the infection by showing, in the same figure, the overall variability in Pseudomonadales levels of the infected and uninfected groups at earlier timepoints and upon clearance (Figure 4D and E). With this panel we are showing the global change of each group which is supported by our statistical analyses shown on Table 3.

Line 261 now reads: “Remarkably, 7/7 t=14 time points converged to the “healthy” microbiome, together with 4/7 T=0 time point samples.”

Figure 3, specify the legend label “infected others” in the text to more clearly state that these are samples collected days 1-7 p.i. from infected ferrets.

Response: The Figure 3 legend has now been modified to reflect the time of collection of the “infected other” samples.

Lines 481-2, can the authors specify how long the animals were held at the institution prior to use in their experiments in the HEPA-filtered incubators? As the authors do not appear to have independently

confirmed that all ferrets were seronegative on day 0, but rather relied on serological data based on pre-shipment potential exposures, it would be beneficial for the reader to know if sufficient time would have elapsed post-shipment where potential exposure to influenza virus (or a different pathogen) may have occurred to explain the divergent profiles observed among 3/7 infected-group ferrets as evidenced in their day 0 microbiome profiles.

Response: We appreciate the methodological clarifications of the animal experiments requested by the reviewer as we agree that this could have an effect on the dynamics of the microbiome. The standard procedure for the work performed with ferret at the Microbiology Department at Icahn School of Medicine is to pre-order influenza free animals from Triple F Farms, which are placed on hold in their specialized facility. To rule out exposure of the animals to influenza, the sera is tested in our laboratory. Only negative animals are then shipped within 24 hrs to our animal facility in specialized transport boxes that are handled only by trained personnel wearing an N95 mask (that prevents the transmission of airborne pathogens). Upon receipt of the animals they are placed immediately individually in the HEPA filtered isolators. Prior to the start of the experiment, the animals are allowed to acclimate for 48 hrs before nasal inoculation. This procedure is performed routinely by our group and other groups, which have included vaccine, challenge and pathology studies. Hence, our animal facility has a number of years of experience using this procedure without suspicions the animals having being exposed to infections during transport or upon receipt to our facility. We have added the requested methodological details to the methods section as suggested by the reviewer:

Lines 478-480: "Upon receipt, the animals were handled only by trained personnel wearing an N95 mask (that prevents the transmission of airborne pathogens) and were immediately housed individually in PlasLabs poultry incubators..."

Lines 482-483: "Prior to the start of the experiment, the animals were allowed to acclimate for 48 hrs before nasal inoculation."

Figure 4, it is still very unclear why the authors are showing microbiome data from an infected ferret that reflects a pre-infection baseline matching <50% of the ferrets in the infection group and <25% of all ferrets in the study. While the authors removed the word "representative" from the main text (yet, neglected to modify the accompanying y axis for panels C and D), it still feels disingenuous to the reader (who is less likely to access the full supplemental data files) to present data in a main display image that is not truly representative of the findings of the study. Ferret 595 shown in Figure 4 appears to have the highest collective peak abundance of pseudomonadales days 5 and 7 p.i. compared with all 7 ferrets in the virus-infected group. Selected data shown in display images should be truly representative of the dataset as a whole, and it does not appear to this reader that the profile for 595 is truly indicative of the group either at day 0 or days 5-7 p.i.

Response: Yes, we accidentally omitted to modify the text on the y axis of Figure 4D and E. We have now corrected this. While we agree with the sentiments of the comment, as discussed in our previous response to the reviewer, the work with outbred ferrets are likely to have a variation in the basal microbiome levels, which is reflected in this study. Of note, in the original review comments the reviewer

already stated that, "The ultimate conclusions drawn by the authors from this data (lines 274-287) that pseudomonadales abundance increases during acute infection, is still supported by the data; the authors just need to be more upfront and revealing about the level of diversity in their challenge experiment so readers can interpret this data fully." Hence, to better depict these results, we believe that the original recommendation of the reviewer, which was to include the data of a second ferret in the main figure, would contribute to the overall appreciation of the diversity and dynamics of the infected group. Hence, we have modified Figure 4 and have now included a second data set from a ferret that has a lower pseudomonadales baseline as suggested by the reviewer and have modified the text accordingly. By including this, the main figure shows now a true representation of the major finding of the study.

Lines 272-284: "composition for two influenza-infected (with divergent baseline microbiomes) and one uninfected ferret (ferret_595 and ferret_587, and ferret_592, respectively) were examined with regards to their taxonomic profiles across six different time points (Fig. 4C-E). At the order level, the IAV-infected ferrets exhibited peak Pseudomonadales abundance at days 5 and 7 dpi (Fig. 4C-G), which correlated with maximal weight loss and peak viral titers (Fig. 4A and B), suggesting the direct or indirect influence of the infection on the microbiome. A phylogenetic inference shows this OTU to be in the order Pseudomonadales but belonging to the genus Acinetobacter (Fig. S5). A few of the less-abundant phyla included Actinobacteria and Firmicutes (Fig. S8). The abundance of Pseudomonadales decreased over time in the infected ferrets, reaching the basal abundance found in healthy ferrets 14 dpi. For the uninfected ferrets, the microbiomes were more stable and Clostridiales was the most abundant taxonomic group, followed by Lactobacillales (light blue). Pseudomonadales were among the least abundant taxonomic group in the uninfected controls (Fig. 4E)."

Reviewer #2 (Remarks to the Author):

Overall the authors have done a thorough job of addressing my comments.

It would be helpful to add, either in the text or in the supplement, a description of how many when the samples were collected from infected human cases. It seems disingenuous to just say multiple samples without specifying the average number per case and showing the distribution for comparison with those taken at specified times from controls.

Response: The information of the samples taken from infected individuals is already described in the following sentences in the text (see below). Additionally, there were few patients with a comprehensive number of timepoints. However, since the samples obtained from infected humans could not be controlled as in the ferrets experiment, there are variations per timepoint per case and we depicted the overall data with a total of 146 swabs collected during the study. We had already noted the following in the text. The total number of samples used for the microbiome analyses was 121. This was specified on line 465.

Additionally, Line 448 states: "Between one and six samples from the acute phase of infection were taken from each patient, together with a sample up to 22 days post diagnosis (convalescence phase or healthy baseline) from most of individuals."

Also, for clarity for the control samples we included the following statement in the methods section:

Line 468: "... ages ranging from 19 year to 65 years..."

Similarly, it would be useful to show the age distributions by age of cases and controls. Looking at the PCAs I did wonder if the controls were a much narrower age range than cases (the table uses a VERY large category (ages 2 to 65) which includes the majority of cases and controls.

***Response:** We are not sure if the reviewer was able to revise Figure S2 which already provides a breakdown of the age range as suggested previously (<2yrs, 2 to 5yrs, 6 to 18yrs, 19 to 64 yrs and 65yrs and older). We have also provided the number of samples for each of those groupings in the figure legend. We do acknowledge that all control cases had ages in the 19-64 range, hence, this is a rather wide age range as represented in the figure. The individual age metadata for all patients is provided in the final data source tables.*

Reviewer #3 (Remarks to the Author):

The authors have adequately addressed all my comments.

***Response:** We appreciate the comments from the reviewer to strengthen the manuscript.*

Reviewers' comments:

Reviewer #1 (Remarks to the Author):

Authors have addressed all comments raised during peer review.

Reviewer #2 (Remarks to the Author):

I realize I did not effectively state my concerns about the analysis of human samples. Observational human studies, such as the one presented here, cannot achieve the same level of control as an experimental study (as noted by the authors), but that does not mean these studies should not be held to a similarly high standard given the limitations.

Several studies have reported that the gut and salivary microbiomes are quite dynamic in early childhood going from less to more diverse. Thus, it seems likely that this is true for the nasopharyngeal microbiome. Just as the authors would not conduct experiments with infected and control samples coming from ferrets known to have different microbiomes it is not scientifically justifiable to have human cases and controls that are known to vary in the diversity and dynamics. Thus, comparing samples from children to adult controls is not scientifically justifiable. Similarly, there are differences among the elderly. There are no controls >65 years. The authors have hid these types of differences in the supplementary material and source code which makes it difficult to evaluate the validity of the human study analysis. For example, they state in line 449 that a convalescent sample was not taken from all individuals, only 'most.' How many individuals did they end up excluding for this reason from the analysis of microbiome dynamics? How many cases are in the same age range of the controls? 2-64 is an enormous age range.

This is also true regarding the number of samples obtained per person. This is because individuals with 6 samples will be more heavily weighted in the analysis than those with only 1 sample. If the authors do the analysis just using two samples from each individual (1 infected and the convalescent sample) are the results the same? Are the samples from the infected period within an individual similar? Is the greatest difference between early infected and the convalescent sample?

I am sympathetic given the time and effort required to conduct human studies, but the authors simply don't have appropriate data to draw the same type of conclusions from the human samples they are from the ferret data.

Point-by-point response to referee's comments: manuscript NCOMMS-19-14037B

Reviewers' comments:

Reviewer #1 (Remarks to the Author):

Authors have addressed all comments raised during peer review.

Response: We appreciate the comments from the reviewer to strengthen the manuscript.

Reviewer #2 (Remarks to the Author):

I realize I did not effectively state my concerns about the analysis of human samples. Observational human studies, such as the one presented here, cannot achieve the same level of control as an experimental study (as noted by the authors), but that does not mean these studies should not be held to a similarly high standard given the limitations.

Several studies have reported that the gut and salivary microbiomes are quite dynamic in early childhood going from less to more diverse. Thus, it seems likely that this is true for the nasopharyngeal microbiome. Just as the authors would not conduct experiments with infected and control samples coming from ferrets known to have different microbiomes it is not scientifically justifiable to have human cases and controls that are known to vary in the diversity and dynamics. Thus, comparing samples from children to adult controls is not scientifically justifiable. Similarly, there are differences among the elderly. There are no controls >65 years. The authors have hid these types of differences in the supplementary material and source code, which makes it difficult to evaluate the validity of the human study analysis.

Response: We appreciate the concerns, but we can confidently say that many of these issues are inconsequential, as addressed and shown in Figure 1. Typically, microbiome ordination plots have more admixture between conditions; here the presence of influenza A virus splits the samples completely across the axis explaining the most variability (PC1). What this means is that we could down-sample and just use one sample from each patient and the IAV+ and IAV- samples would still cluster separately with statistical significance.

Similarly, we acknowledge the reasoning of the reviewer that children less than 5 years of age might have different and developing microbiomes. In this study only four children are < 5 years of age (clearly shown in Figure S2 A, orange and yellow dots). The PCoA plots displayed on Figure 1 and Figure S2 (included in the first round of reviews) and Table S3, show very conclusively that time points, age, and vaccination status, among other factors, had little influence on any real grouping of the samples (e.g. samples of the same are distributed homogeneously within the cluster). Instead, as mentioned above, the ordination plots separate the groups relative to whether the sample comes from an IAV infected individual or not. Hence, along the same lines, this also indicates that including negative controls >65 years of

age would provide no additional statistical power to the analyses given that infection status and not age, drives the clustering and the Ecostates (Table 2) with high statistical support.

For example, they state in line 449 that a convalescent sample was not taken from all individuals, only ‘most.’ How many individuals did they end up excluding for this reason from the analysis of microbiome dynamics?

Response: The relevant text highlighted by the reviewer is “Between one and six samples from the acute phase of infection were taken from each patient, together with a sample up to 22 days post diagnosis (convalescence phase or healthy baseline) from most of individuals.”

Specifically, samples from 14 out 28 individuals were collected during the convalescent state or healthy state. To clarify the misunderstanding of the reviewer, the convalescent or healthy state samples of the reminder 14 individuals were not excluded, instead these samples were not obtained because these individuals could not be recruited back for that visit. Nevertheless, in all the microbiome analyses, including the analyses of microbiome dynamics, we included the samples from all time points from all the patient (i.e. no samples were excluded in any of the analyses, except for the samples that were not confirmed to have been infected with influenza virus, which is stated in the methods, lines 441-445). Moreover, due to the comprehensive analyses conducted, including additional samples from day 22 would not change the results. If anything, in the scenario that we were able to get those samples, we might have seen a few more microbiomes recovering to the healthy state, further supporting the findings reported in the manuscript.

For clarification, the text now reads (lines 447-450): “Between one and six samples from the acute phase of infection were taken from each patient, together with a sample up to 22 days post diagnosis (convalescence phase or healthy baseline) from 14 out of the 28 individuals analyzed.”

How many cases are in the same age range of the controls? 2-64 is an enormous age range.

Response: From Table S1, there are 17 cases in the 2-64 age range. We appreciate that this is a large range, but finer granularity (i.e. more age categories) would dilute statistical power. In addition, Figure 2S A shows that individuals of the same age range don’t particularly cluster together.

This is also true regarding the number of samples obtained per person. This is because individuals with 6 samples will be more heavily weighted in the analysis than those with only 1 sample. If the authors do the analysis just using two samples from each individual (1 infected and the convalescent sample) are the results the same? Are the samples from the infected period within an individual similar? Is the greatest difference between early infected and the convalescent sample?

Response: Please see note above... we could use 1 sample from each patient and would still get clear microbiome separation between IAV+ and IAV- samples.

I am sympathetic given the time and effort required to conduct human studies, but the authors simply don't have appropriate data to draw the same type of conclusions from the human samples they are from the ferret data.

Response: We appreciate the reviewer's consideration of the difficulties and challenges of conducting human studies. In fact, while addressing comments from Reviewer #3 in the first rounds of reviews we already acknowledged some of the limitations in the analyses of the human samples. Then we stated that "it would be an extremely complex and expensive clinical set up to obtain an ideal control cohort", and that we "believe there is not an ideal control for the human infection study, and that is why we performed the complementary ferret study." Our comprehensive analyses and results argue strongly that the effect of clustering of microbiomes of the human samples is due to infection states and not other factors (see responses above), and that this change in microbiome structure occurs in a dynamic manner. The ferret data demonstrates the same phenomenon of clustering by Ecostates, and shows a dynamic change in microbiome during influenza infection. Consequently, the data presented strongly supports these conclusions from both human and ferret samples. Differences in the diagnostic OTU of the Ecostates in both species are reported, and hence those conclusions are thus different and have been clearly stated.